# Measuring error rates in genomic perturbation screens: gold standards for human functional genomics

Traver Hart[1], Kevin R Brown[1], Fabrice Sircoulomb[2], Robert Rottapel[2,3,4] & Jason Moffat[1,5,*]

## Abstract

**Technological advancement has opened the door to systematic genetics in mammalian cells. Genome-scale loss-of-function screens can assay fitness defects induced by partial gene knockdown, using RNA interference, or complete gene knockout, using new CRISPR techniques. These screens can reveal the basic blueprint required for cellular proliferation. Moreover, comparing healthy to cancerous tissue can uncover genes that are essential only in the tumor; these genes are targets for the development of specific anticancer therapies. Unfortunately, progress in this field has been hampered by off-target effects of perturbation reagents and poorly quantified error rates in large-scale screens. To improve the quality of information derived from these screens, and to provide a framework for understanding the capabilities and limitations of CRISPR technology, we derive gold-standard reference sets of essential and nonessential genes, and provide a Bayesian classifier of gene essentiality that outperforms current methods on both RNAi and CRISPR screens. Our results indicate that CRISPR technology is more sensitive than RNAi and that both techniques have nontrivial false discovery rates that can be mitigated by rigorous analytical methods.**

**Keywords** cancer; CRISPR; essential genes; RNAi; shRNA
**Subject Categories** Methods & Resources; Chromatin, Epigenetics, Genomics & Functional Genomics
**Mol Syst Biol. (2014) 10: 733**

See also: **B Evers *et al*** (June 2014)

## Introduction

In the early 1900s, Lucien Cuénot observed unusual patterns of inheritance when studying coat color in mice and, from his many crosses, never produced a single homozygous yellow mouse (Cuenot, 1905; Paigen, 2003). Not long after these observations, it was shown that Cuenot's crosses resulted in what appeared to be non-Mendelian ratios because he had discovered a lethal gene (Castle & Little, 1910). W.E. Castle and C.C. Little demonstrated that one-quarter of the offspring from Cuénot's crosses between heterozygotes died during embryonic development, ushering in embryonic lethality, or death, as a new phenotypic class for geneticists (Castle & Little, 1910). Consequently, the idea that organisms harbor sets of lethal or essential genes has taken shape over the past century. In the past dozen years or so, systematic genomic studies in eukaryotic model systems have defined sets of lethal or essential genes under defined growth conditions, providing a nexus for biologists to study the essential molecular processes that occur during cell growth and proliferation.

The importance of defining essential genes is threefold. First, it provides a blueprint for all components necessary for a cell to grow and divide under defined conditions. Second, it provides a parts list that can be deconstructed to uncover all the necessary cellular and molecular functions that proceed during cell growth and division under defined experimental conditions. Third, the list of essential genes and related functions provides a reference point for understanding disease. Indeed, the accurate identification of human disease genes is among the most important goals of biomedical research, and there exists a complex relationship between disease genes and essential genes, particularly for cancer genomes. For example, a recent analysis has shown that the cumulative effects of copy number variants of cancer drivers and essential genes along a chromosome explain the recurring patterns of somatic copy number alterations of whole chromosomes and chromosome arms in cancer genomes (Davoli *et al*, 2013).

Broadly speaking, a gene is defined as essential if its complete loss of function results in a complete loss of fitness. In single-celled organisms, this is a fairly straightforward assessment; however, in metazoans, a gene could be reasonably classified as essential if its loss of function resulted in sterility or failure to develop to adulthood. In practice, a prenatal lethal phenotype is

1   Donnelly Centre and Banting and Best Department of Medical Research, University of Toronto, Toronto, ON, Canada
2   Campbell Family Cancer Research Institute, Ontario Cancer Institute, Princess Margaret Hospital, University Health Network, Toronto, ON, Canada
3   Department of Medical Biophysics, University of Toronto, Toronto, ON, Canada
4   Division of Rheumatology, Department of Medicine, St. Michael's Hospital, Toronto, ON, Canada
5   Department of Molecular Genetics, University of Toronto, Toronto, ON, Canada
    *Corresponding author. Tel: +1 416 978 4019; E-mail: j.moffat@utoronto.ca

typically the criterion for essentiality. Given the absence of a set of well-established human essential genes, researchers have generally relied on orthology to infer essentiality. Lethal or essential gene sets have been generated under defined growth conditions for a number of eukaryotic model systems including the budding yeast *S. cerevisiae* (Winzeler *et al*, 1999; Giaever *et al*, 2002), the fission yeast *S. pombe* (Kim *et al*, 2010), *C. elegans* (Kamath *et al*, 2003), *D. melanogaster* (Boutros *et al*, 2004; Dietzl *et al*, 2007), *M. musculus* (White *et al*, 2013) and others. Across model organisms, essential genes are more likely to be hubs in protein-protein interaction networks (Jeong *et al*, 2001), a phenomenon driven to some degree by membership in large essential protein complexes (Hart *et al*, 2007; Zotenko *et al*, 2008). Moreover, model organism essentials are less likely to have paralogs (Makino *et al*, 2009), consistent with the model of gene duplication buffering loss-of-function phenotypes (Gu *et al*, 2003). Human orthologs of mouse knockouts which give rise to developmental lethal phenotypes are themselves enriched for developmental disease genes, even above the bias toward developmental genes in the mouse knockout set (Makino *et al*, 2009). Furthermore, ubiquitously expressed human genes are very likely to contain a large proportion of essential genes and are different in their evolutionary conservation rates (i.e. higher nonsynonomous/synonomous substitution rates), DNA coding lengths, and gene functions compared with disease genes and other genes (Tu *et al*, 2006).

Experimental assays of human gene essentiality are performed in cell lines. RNA interference has, to-date, been the weapon of choice for genome-scale fitness screening, with roughly two hundred published cell line screens (Luo *et al*, 2008; Schlabach *et al*, 2008; Silva *et al*, 2008; Cheung *et al*, 2011; Marcotte *et al*, 2012). Other approaches include gene traps in haploid human cells (Carette *et al*, 2009; Burckstummer *et al*, 2013) and, more recently, genome-scale gene editing approaches using lentiviral-based CRISPR technologies (Shalem *et al*, 2013; Wang *et al*, 2013). The RNAi screens to-date have typically been conducted in cancer cell lines or normal counterparts to elucidate not only which genes are essential, but also which genes are differentially essential in different contexts, with the ultimate goal of identifying genes or pathways that are tissue-, subtype-, or even tumor-specific (i.e. genotype- or context-dependent essential or lethal genes). With the widespread adoption of pooled library shRNA screens has come the understanding that there are caveats to this type of genetic screening approach. In particular, off-target effects can lead to false positives (Echeverri *et al*, 2006; Moffat *et al*, 2007), if the unintended target of an shRNA hairpin is an essential gene. To mitigate these effects, analytical approaches have been developed that look for phenotypic consistency across multiple hairpins targeting a gene (Luo *et al*, 2008; Cheung *et al*, 2011; Marcotte *et al*, 2012) and among the same hairpins in different screens (Shao *et al*, 2013). Not surprisingly, different approaches can yield different results, and the degree to which false positives contaminate results is largely unknown (Kaelin, 2012).

No method currently exists to systematically evaluate these various approaches. Studies in other areas of functional genomics have relied on 'gold-standard' positive and negative reference sets (Jansen & Gerstein, 2004) to evaluate the sensitivity and specificity of, for example, protein-protein interactions (Hart *et al*, 2006; Havugimana *et al*, 2012). This approach applies equally well to gene essentiality studies, where negatives can outnumber positives

by an order of magnitude. However, no such gold standards currently exist for screens using mammalian, and more specifically human, cell lines. The developmental essentials inferred by orthology contain many genes which are, by definition, essential for whole-organism development but unlikely to be essential in any given cell line context. To our knowledge, no putative cell line nonessential reference set exists at all, though it is certainly an impossible task to prove that any gene is nonessential in all contexts.

In this study, we derive gold-standard reference sets of human cell line essential and nonessential genes. We use them to train a Bayesian classifier of gene essentiality in pooled library shRNA screens and, most importantly, to evaluate the error rates of individual screens. We demonstrate how to leverage this framework to evaluate the data quality of genome-scale fitness screens in human cell lines as well as the effectiveness of the analytical approaches applied to them. In addition, we develop models of gene essentiality that permit estimation of the number of core essential genes and total number of essential genes. Our method is applicable to new pooled screening methodologies such as gene traps with haploid cell lines (Carette *et al*, 2009) or genome-scale pooled CRISPR approaches (Shalem *et al*, 2013; Wang *et al*, 2013). The reference sets can be used to evaluate screen quality regardless of what analytical method is applied.

# Results

## Computational framework for predicting essential genes from reverse genetic screens

Genetic screens in mammalian cells using pooled barcoding approaches have emerged as a powerful method for functional discovery. In particular, negative genetic selections have the potential to reveal entire genetic pathways that govern cell growth and proliferation (i.e. essential/lethal factors). In order to advance our ability to analyze and assess the quality of systematic genetic screens that are emerging, we developed an informatics approach that is applicable to any genome-scale genetic screen or set of screens for predicting essential/lethal factors. Using a compendium of shRNA screens across different human cancer cell lines (Marcotte *et al*, 2012), we developed a Bayesian classifier to score essential/lethal factors. As cells harboring shRNA hairpins targeting essential/lethal factors drop out of a proliferating population, the corresponding shRNAs show strong negative fold-change relative to controls. The data for each cell line are comprised of microarray data for up to three replicates at an initial timepoint (T0) and each of two experimental timepoints, and we calculated fold-change for each observation relative to the mean of the control microarrays, resulting in a matrix of fold-changes for ~78,000 hairpins across nearly 400 cell lines/timepoints. The Bayesian classifier was developed to evaluate whether the distribution of fold-changes for hairpins targeting a given gene most closely matched the distribution of fold-changes of hairpins targeting training sets of essential genes or nonessential genes using twofold cross-validation to prevent circularity (Fig 1). The classifier was trained on reference sets we generated, and each screen's performance was evaluated against a withheld test set (Fig 1).

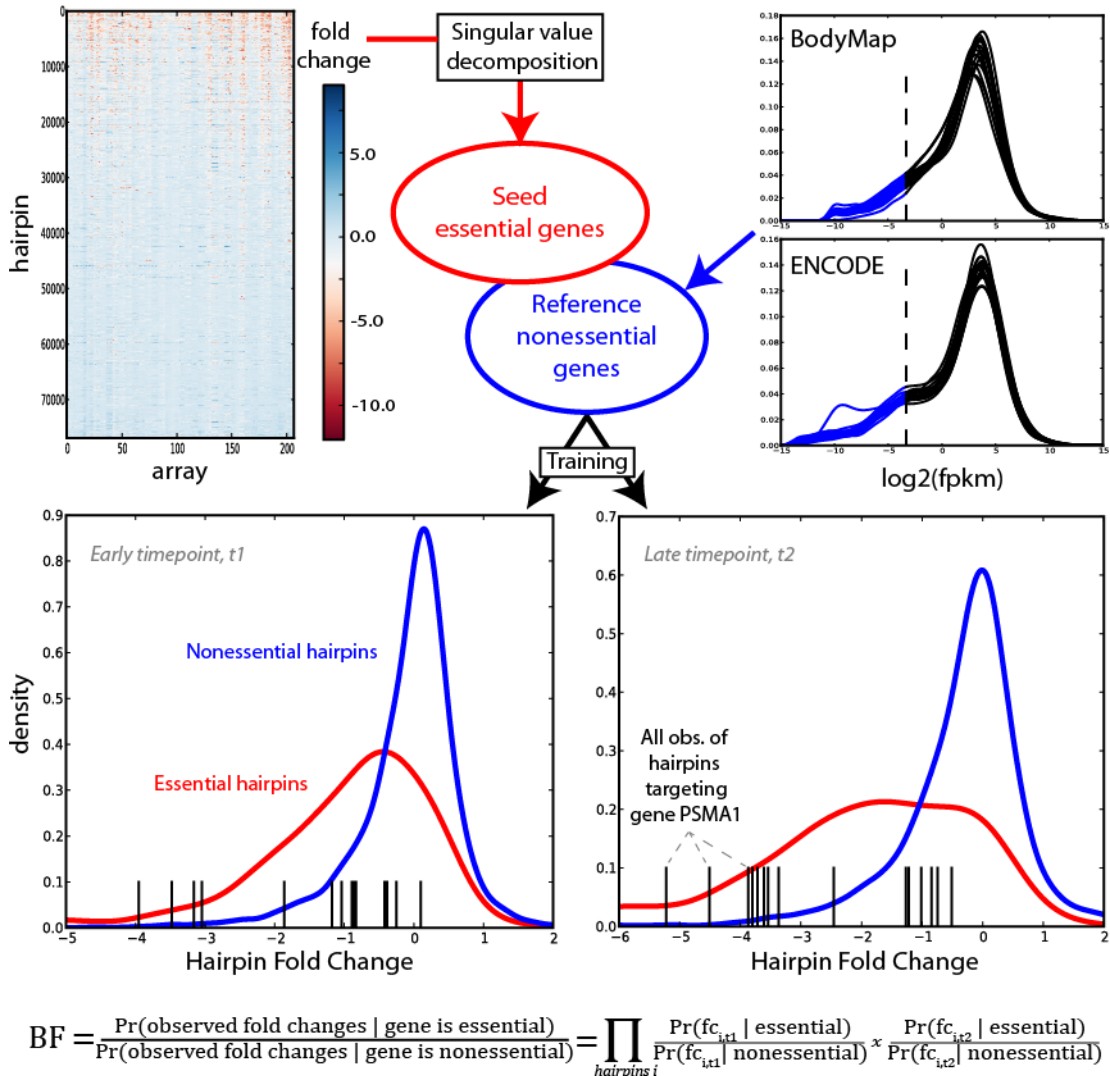

$$BF = \frac{Pr(\text{observed fold changes} \mid \text{gene is essential})}{Pr(\text{observed fold changes} \mid \text{gene is nonessential})} = \prod_{hairpins\ i} \frac{Pr(fc_{i,t1} \mid \text{essential})}{Pr(fc_{i,t1} \mid \text{nonessential})} \times \frac{Pr(fc_{i,t2} \mid \text{essential})}{Pr(fc_{i,t2} \mid \text{nonessential})}$$

**Figure 1. Analytical overview.**
Half of the matrix of shRNA hairpins was decomposed using linear algebra techniques to find a set of reference essential genes. Reference nonessentials were derived from low-expression genes across a compendium of RNA-seq experiments. For each cell line/timepoint in the 2nd half of the shRNA data, the empirical distributions of training essentials and nonessentials were determined, and for each remaining gene, a Bayes Factor (BF) is calculated which measures which distribution its cognate hairpin data most closely matches.

*Source of reference essential genes (EGs)*
An effective reference set of EGs should include all genes that are essential across every cell line or context in which they have been studied. We used a linear algebra approach to find genes that were consistently essential across cell line screens previously performed in our laboratory (Marcotte *et al*, 2012). Singular value decomposition (SVD) is a matrix factorization technique that yields a set of orthogonal basis vectors that describe, in rank order, the major sources of variation in the data. Briefly, SVD was applied to half of the shRNA fold-change matrix, yielding one dominant left singular vector (LSV) that describes ~42% of the total variance in the matrix (Supplementary Fig S1A). The distribution of all shRNA projections onto this first LSV is shown in Supplementary Fig S1B. shRNAs with strong positive projections show consistent dropout across effective shRNA screens, which had strong negative projections

onto the corresponding right singular vector (Supplementary Fig S1C), projections which correlated with the number of cell doublings at which each sample was measured (Supplementary Fig S1D). For each gene, we found the median projection onto the first LSV of its cognate hairpins and measured whether hairpins with median rank or higher rank were enriched in the right tail by a hypergeometric test, yielding 179 genes at a false discovery rate (FDR) of 25% (Benjamini & Hochberg). This list was further filtered to 148 constitutively and invariantly expressed genes, that is, genes with mean $\log_2(\text{FPKM}) > 0$ across both the ENCODE [17 cell lines (Tilgner *et al*, 2012)] and Illumina BodyMap (16 tissues, EBI accession no. E-MTAB-513) RNA-seq datasets, and standard deviation of $\log_2(\text{FPKM})$ less than the mean standard deviation of all observed protein-coding genes in each dataset (Supplementary Fig S2).

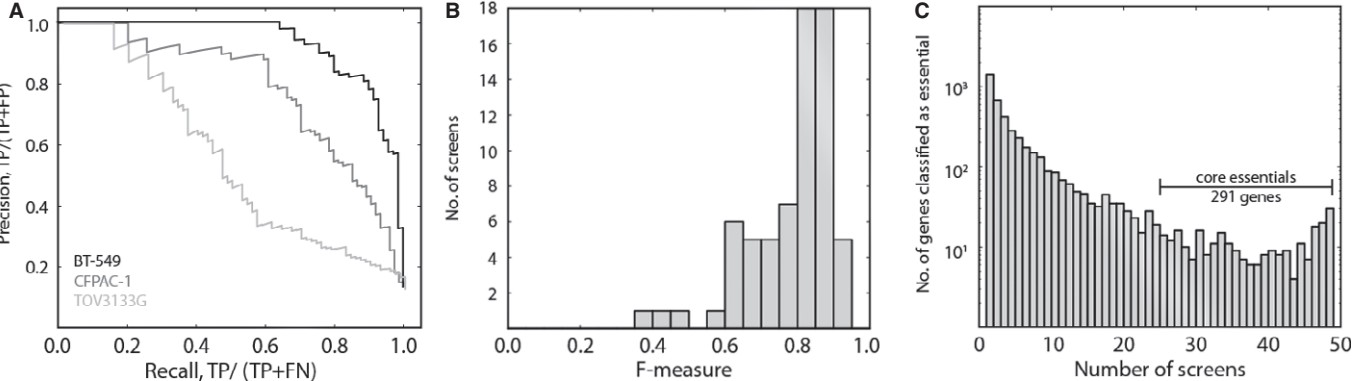

**Figure 2. Screen quality and core essentials.**

A For each screen, genes are ranked by BF and evaluated against a test set of reference essentials and nonessentials, and a precision vs. recall (PR) curve is calculated. Three screens representing the variability in global performance are shown.

B Distribution of *F*-measures of the 68 screens used in this study. Screens with *F*-measure > 0.75 (*n* = 48) were considered high-performing and were retained for downstream analyses.

C Histogram of essential gene observations across the 48 performing cell lines. Genes essential in 24/48 lines (*n* = 291) were considered core essentials. Genes observed in only 1–3 cell lines are highly enriched for false positives.

## Source of reference nonessential genes (NEGs)

Defining a reference set of nonessential genes is less clear cut, as it is impossible to experimentally demonstrate nonessentiality in all contexts. However, we reasoned that genes that are not expressed in the majority of tissues and cell lines are reasonable candidates for such a reference set. To generate this set, we again turned to published RNA-seq data. We selected protein-coding genes that are probed by our shRNA library and have an expression level of less than 0.1 FPKM in 15 of 16 BodyMap tissues and 16 of 17 ENCODE cell lines, as genes expressed below this level are typically not biologically relevant (Hebenstreit *et al*, 2011; Hart *et al*, 2013). We label the resulting set of 927 putatively nonessential genes the *NEG* set. While this set may include some genes that are essential in other cellular or organismal contexts, the net effect of a small number of 'accidental essentials' in this set should be negligible. The seed and reference nonessential genes are listed in Supplementary Dataset S1.

## Bayes Factor scores

Reference essentials and nonessentials were divided into equal-sized training and testing sets for subsequent analyses, and each cell line in the withheld half of the shRNA fold-change matrix was analyzed independently. For each timepoint, the fold-change distributions for the essential and nonessential training sets, comprising 347 and 2,268 hairpins respectively, were determined. Then, for each gene, a Bayes Factor (BF) was calculated, representing the log likelihood that the observed fold-change for a given gene's cognate hairpins was drawn from either the essential or the nonessential reference distribution. Log BFs were summed across all time points for a final BF for each gene in each cell line. Supplementary Dataset S2 contains a table of all calculated Bayes Factors.

## F-measure

For each cell line, genes were rank-ordered by BF and compared to the withheld reference test sets to evaluate precision vs. recall.

Screen quality varied widely, with most screens showing moderate to high performance, though several outliers showed remarkably poor results (Fig 2A). We identified the point on the recall-precision curve for each screen where the BF crossed zero and calculated the *F*-measure (harmonic mean of recall & precision) of each screen at that point. We judged screens with *F*-measure ≥ 0.75 (*n* = 48/68; Fig 2B) to be high-performing screens and retained them for downstream analyses. Screen performance measures are listed in Supplementary Dataset S3.

## Core essentials

Within this set of high-performing screens, we examined the frequency with which each gene was called essential (BF > 0) (Fig 2C). Though 4,451 genes have a positive BF in at least one cell line, genes observed in few (1–4) screens are enriched for false positives. Repeated observation greatly improves the likelihood that a gene is truly essential. To identify likely global essential genes, and to avoid identifying cancer tissue/subtype-specific genes, we selected genes observed in at least half of the performing screens (n = 291 genes). We label these *core essentials*.

### Cumulative analysis of EGs

To identify the set of all EGs observed across all screens, we used a cumulative analysis approach. Most large-scale functional genomics screens try to differentiate a small number of true 'hits' from a pool of negatives that can often be orders of magnitude larger (Jansen & Gerstein, 2004). In such screens, even a tiny false positive rate applied across all true negatives can result in large FDRs for individual screens.

Researchers can attenuate the final FDR by conducting multiple repeats of screens and analyzing the frequency with which each hit is observed across repeats. By considering the cumulative distribution of hits across multiple screens, information about both the total number of true essentials in the population and the error rate in observing those hits can be calculated. In principle, a screen with

 

zero FDR that is repeated to saturation will yield a cumulative observations curve that flattens to a slope of zero at the total number of hits in the screened population. In practice, repeated screens can saturate the true hits, but random discovery of false positives yields a cumulative curve with positive slope as more and more false positives are accumulated. A variation of this cumulative observation analysis was used to evaluate the saturation of protein-protein interaction screens (Hart *et al*, 2006).

To test this logic, we rank-ordered the top-performing 36 cell line screens in our compendium of pooled shRNA screens by *F*-measure and plotted the cumulative number of observed essential genes. The result is indeed a curve that flattens but with a positive slope (Fig 3A). To estimate both the number of essential genes and the average screen error rate, we conducted *in silico* simulations of the 36 screens, determined the synthetic cumulative observation curve for each set of simulations, and measured the curve's fit to our experimental observations. With fixed parameters of 15,687 genes assayed and 606 genes reported as essential in each screen (the mean number of genes in the top 36 screens with BF > 0), we find that a model with a cellular population of 1,025 essential genes and an average screen FDR of 15% yields a cumulative essentials curve that mimics the observed curve very closely (Fig 3A). Running the model across a range of total essential population sizes and FDRs and calculating root-mean-squared deviation (RMSD) from the observed cumulative essentials curve show models with 850–1,175 essential genes, and FDRs of 14.0–16.5% yield an RMSD that is less than 1.5× the minimum RMSD (Fig 3B). Notably, the FDR range for the best fit models is highly consistent with the average empirically measured FDR of 13.8% across the top 36 screens. Moreover, while the top screens encompass several cancer subtypes from three tissues of origin, the model treats all 36 repeats as replicates. Tissue- or subtype-specific essential genes in the saturated region will be incorrectly treated as false positives using the cumulative approach; therefore, FDR estimates derived in this manner are likely conservative.

Though the modeling approach can tell us approximately how many essential genes are in our cell lines *in toto*, it does not identify which genes are truly essential. To separate essential genes from false positives, we rely on repeat observations of essential genes across multiple screens. Fig. 3C shows a histogram of gene essentiality calls across the top 12 screens. Of the 2,130 unique hits in these screens, 945 (44%) are observed in a single screen, while only 392 (18%) are seen in six or more of the 12 cell lines.

To estimate the binwise FDR for this distribution, we again turn to the cumulative approach. Fig 3C also shows the distribution of essential gene calls in the 13th–24th ranked screens (blue) and the 25th–36th ranked screens (red). If we assume that the first 12 screens have achieved saturation—a likely false assumption but a useful approximation for modeling—then all subsequent hits must be false positives. The second and third sets of 12 screens therefore model the frequency distribution of false positives and give an estimate of the expected number of false positives in each bin. Based on these estimates, we conclude that hits in 3 or more of the top 12 screens are essential genes with an FDR of 6–11%. This set comprises 823 genes, which we label *total essentials* (see Supplementary Dataset S4 for a complete list), and contains all 291 core essentials.

Given the diversity of tissues and subtypes in the cell lines studied, it is highly unlikely that all observations beyond the top 12 screens are false positives. Some fraction of subsequent hits may in fact be true subtype-specific essential genes. For example, the top 12 cell lines include five pancreatic, three ovarian and four breast cancer cell lines, of which three are basal subtype and one, HCC-1954, is EGFR-high/Her2 amplified. Well-studied subtype-specific breast cancer oncogenes CDK4 and FOXA1 are not classified as essential in any of the top 12 screens, including HCC-1954, though this line does show a dependence on Her2/ERBB2 (BF = 9.21). However, across all 48 performing screens, CDK4 and FOXA1 each show BF > 20 in 4 cell lines; two of the four CDK4 lines and three of the four FOXA1 lines are HER2[+] breast cancer lines, and the remainder are all luminal subtype. The net effect of these subtype-specific essentials in the analysis of cumulative observations is to artificially inflate the imputed number of false positives in each bin, thus rendering our FDR estimates conservative.

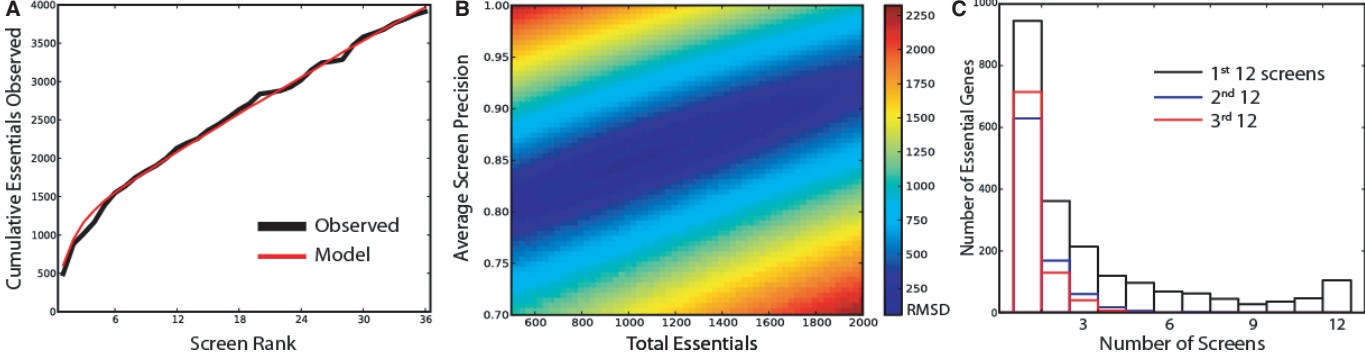

**Figure 3.  The cumulative model of essential genes.**

A    The top 36 cell lines were rank-ordered by *F*-measure, and the cumulative count of classified essential genes was plotted (black curve). Simulated repeat experiments sampling a population of 1,025 essential genes at 15% FDR yield a similar cumulative count (red curve).

B    In simulated repeat experiments across parameter space, models sampling 875–1,175 essential genes at 13.5–16.5% FDR (1-Precision) yielded cumulative observation curves similar to what was observed experimentally.

C    Histogram of observations of essential genes in top-ranked 12 screens (black), genes exclusive to the next set of 12 (blue), and exclusive to the 3rd set of 12 (red). Genes observed in at least 3 of the top 12 screens are classified as global essentials.

### Characteristics of EGs

Core essential genes are expected to be essential in all cell lines and contexts and must be constitutively expressed. Indeed, 217 of 291 core essentials (74.6%), as well as 483 of 823 total essentials (58.7%), showed high mean expression with low variation across a compendium of RNA-seq experiments (Supplementary Fig S2), compared to 33.4% of other genes. These genes are also highly enriched for protein complexes; more than half of the set of total essentials encode subunits of annotated human protein complexes. Fig 4A shows the top nonoverlapping (i.e. minimal shared subunits) protein complexes that show strong enrichment for essential genes (comprising 231 genes), with most subunits detected as core essentials and coverage increased by the set of total essentials (see Supplementary Dataset S5 for a complete list). These complexes

represent the fundamental molecular functions of cellular life: transcription, translation, and replication. An additional 235 essential genes are also annotated as subunits of protein complexes, though the complexes do not meet our threshold for statistical significance. The remaining 357 essential genes not in any annotated protein complex also show enrichment for core cellular processes, including ribosome biogenesis (13 genes, 5.6-fold enrichment, $P = 3.6e-6$), aminoacyl-tRNA synthetases (4 genes, 6.1-fold, $P = 2.8e-2$), and protein tyrosine phosphatases (8 genes, 4.3-fold, $P = 2.7e-3$). Essential genes not in complexes are generally not constitutively expressed; 117 of the 357 (32.8%) show constitutive and invariant expression compared to 33.4% of nonessentials, suggesting this may be a rich source of tissue-specific essentials.

Essential genes were divided into the categories described above (i.e. in enriched complexes, in other complexes, not in any complex;

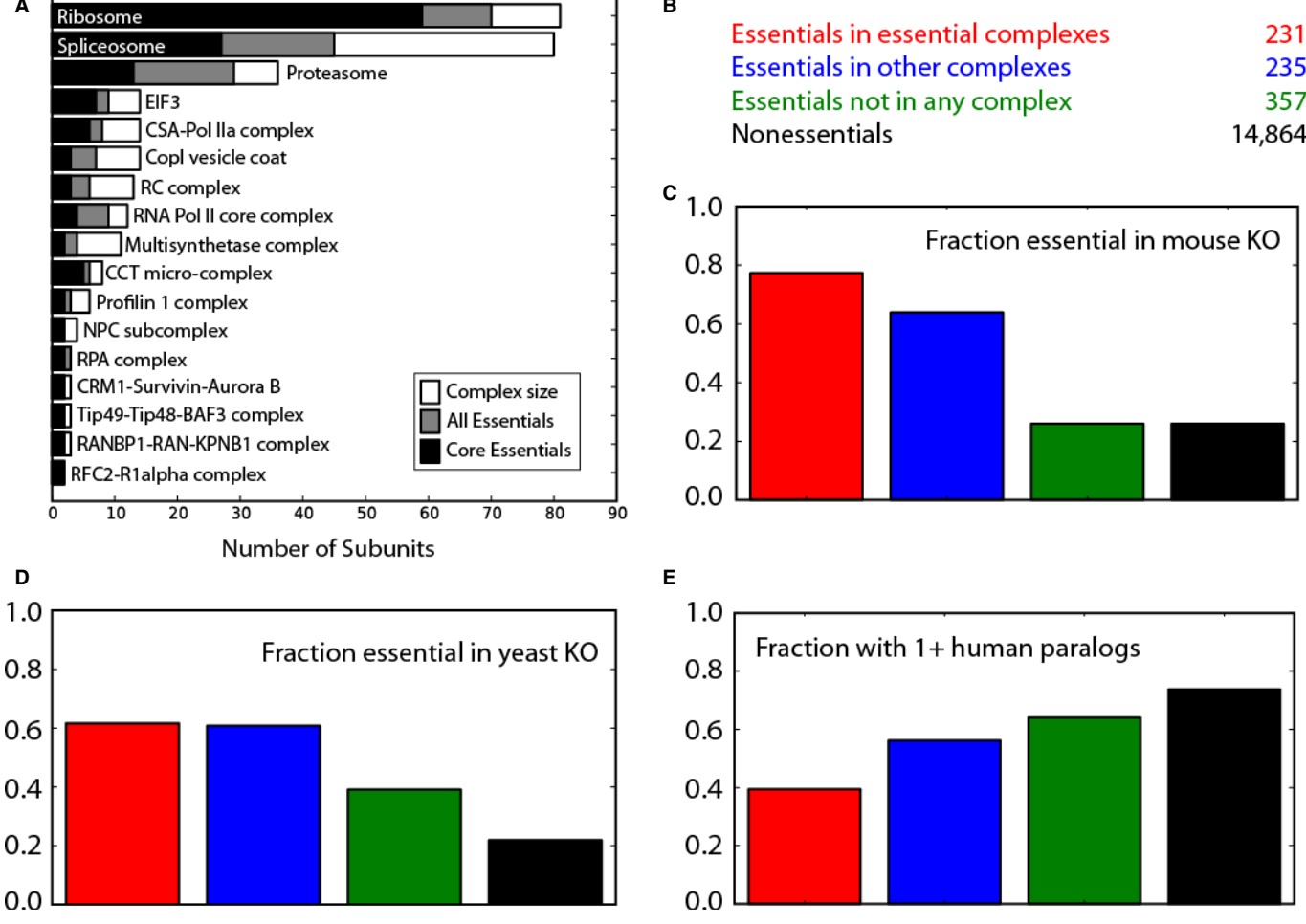

**Figure 4.  Characteristics of essential genes.**

A  Essential genes are highly enriched for core protein complexes. Seventeen representative nonoverlapping complexes are shown, with the core essentials (black) and total essentials (gray) shown relative to the total number of subunits in the complex.

B  Total essentials are separated into categories: those in complexes enriched for essential genes, those in other complexes but which fail enrichment tests, and those not annotated to be in any protein complex. The remaining genes are classified as nonessential.

C  Fraction of genes in each category whose mouse orthologs are also essential; colors as in (B).

D  Fraction of genes in each category whose yeast orthologs are also essential; colors as in (B).

E  Fraction of genes in each category with one or more human paralogs; colors as in (B).

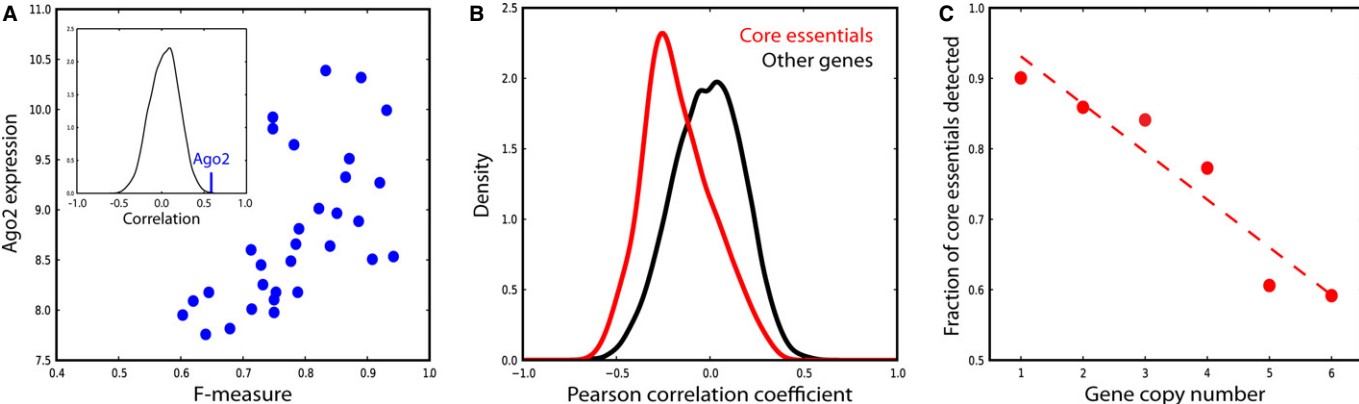

**Figure 5.  Biological drivers of variation in RNAi screen efficacy.**

A   Plotting Ago2 gene expression (measured by microarrays; *y*-axis) versus cell line *F*-measure (*x*-axis) for pancreatic, and ovarian cancer cell lines reveals strong correlation (Pearson's *r* = 0.59). Inset, distribution of correlations of expressed genes (*n* = 10,673) versus *F*-measure; Ago2 is the top-ranked gene.

B   The Pearson's correlation coefficient of absolute copy number vs. Bayes Factor was determined for all genes across 30 pancreatic and ovarian cancer cell lines. Core essential genes show a negative correlation between copy number and essentiality.

C   Core essential genes were binned by absolute copy number across the 30 samples. In each bin, the fraction of core essentials that were accurately classified in the corresponding screens is plotted. High copy number among core essentials reduces sensitivity to RNAi.

Fig 4B) to examine the fraction of genes in each of these categories that overlap/intersect with mouse essential genes or represent paralogs. For genes whose mouse orthologs have been knocked out, essentials in protein complexes were much more likely to have an essential mouse or yeast ortholog than other genes (Fig 4C and D). Furthermore, we find that essential genes in protein complexes are less likely to have paralogs than nonessential genes (Fig 4E). In particular, essential genes in essential complexes are more likely to be singletons than other classes (Fig 4E).

**Biological sources of variability in RNAi negative selection screens**

Having derived a set of performance metrics using essential genes at the screen and gene level, we sought to understand some of the drivers of variability, particularly in lentiviral-based pooled RNA interference screens across a large panel of human cancer cell lines. Fortunately, the pancreatic and ovarian cancer cell line screens have matching gene expression microarray data collected on the same array platform (Marcotte *et al*, 2012). Measuring the correlation between gene expression and screen *F*-measure across 31 cell lines (one outlier removed), we found that AGO2 had the top-ranked correlation among more than 10,000 expressed genes (Pearson's correlation coefficient = 0.59; Fig 5A). The AGO2 protein, coupled with short RNA, comprises the RNA-induced silencing complex (RISC), which catalyzes the cleavage of target mRNA and was expected to be an important predictor of RNAi efficiency. The relationship between AGO2 mRNA expression and shRNA screen quality was weaker in the breast cancer screens (Supplementary Fig S3), which may reflect some combination of generally better performing screens in breast cancer cell lines—with corresponding lower variability—and the fact that the expression data were collected on a different microarray platform.

While AGO2 expression may help explain why some screens perform better than others, it does little to explain the variability within high-performing screens. Though the large number of genes observed infrequently in Fig 2C reflects the expected distribution of false positives across the screens, we expected a more pronounced peak at the right edge of the distribution from core essentials observed across most or all high-performing screens. We explored other molecular genetic data to explain this false negative rate among known essentials and derived absolute copy number for each gene across 30 pancreatic and ovarian cancer cell lines in our study (see Materials and Methods and Supplementary Dataset S6). We calculated a Pearson's correlation coefficient for each gene's copy number profile vs. its Bayes Factor profile across the same screens and observed that core essential genes show a negative correlation between copy number and essentiality (Fig 5B). Notably, the core essential genes largely encode members of essential protein complexes, and our observation is consistent with a model whereby increased copy number yields protein levels in excess of stoichiometric requirements for protein complex function. The genes are likely no less essential, as complete knockout would probably still kill the cells, but the copy number amplification renders them less sensitive to RNAi perturbation. Binning core essential genes by absolute copy number and measuring the fraction in each bin that are successfully identified in the screens (Fig 5C) support this model. In other words, as copy number increases, the likelihood that a core essential is accurately classified drops markedly. Based on the difference between the overall observed false negative rate and the false negative rate at copy number = 2, we estimate that 15–20% of false negatives (core essentials not accurately classified as essential in a cell line) are attributable to copy number variation. This hypothesis could be tested by employing orthogonal genome-editing technologies, such as CRISPR, though such technologies might also be limited.

**Leveraging gold-standard reference sets to improve analyses of CRISPR and shRNA screens**

We used matrix decomposition to generate a seed set of reference global essentials to train our Bayesian classifier, the application of

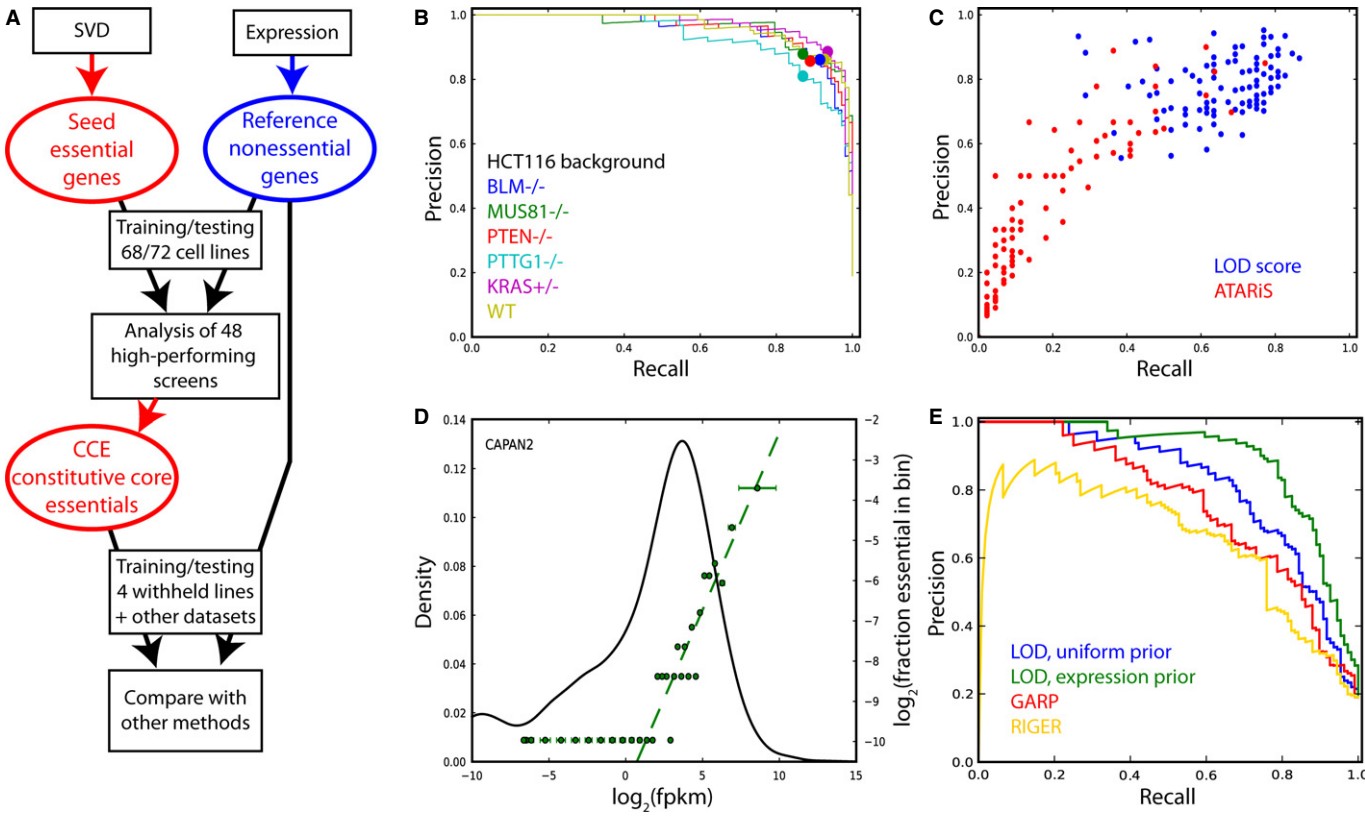

**Figure 6. Evaluating other shRNA data and methods.**

A   Analytical approach. CCE reference set was derived from the initial analysis; NE set is identical throughout.

B, C   Evaluating other RNAi data sets. (B) LOD scores were calculated for the pooled library shRNA screens in the HCT116 background in (Vizeacoumar *et al*, 2013) and evaluated against CCE-test and NE-test. Recall, TP/(TP+FN); Precision, TP/(TP+FP). All six screens showed very high accuracy. The filled circle indicates the point on the curve where LOD = 0. (C) LOD scores were calculated for the pooled library shRNA screens in 102 cancer cell lines in (Cheung *et al*, 2011). Blue points represent recall & precision at LOD = 0 as measured against CCE-test and NE-test. Red, recall and precision for the same cell lines and same reference sets from ATARiS gene solutions at phenotype score = −1.

D   Integrating gene expression into the Bayesian classifier. For RNAi screens with matched gene expression data (in this example, PDAC cell line CAPAN-2, black curve), genes are binned by expression level and the fraction of reference essentials in each bin (right *y*-axis) is plotted against the mean expression of genes in the bin (green points). A linear fit on the log–log plot (green dashed line) can be integrated into the Bayesian classifier as an informative prior.

E   Integrating expression data improves the performance of the classifier (green) over the base algorithm (blue). Both forms show better performance than other algorithms such as GARP (red) and RIGER (gold).

which ultimately yielded 291 core essential genes across 48 high-performing shRNA screens of 68 cell lines. We filtered these genes for constitutive, invariant expression across the BodyMap and ENCODE RNA-seq samples, yielding a set of 217 genes we label Constitutive Core Essentials (*CCE*). We then divided the *CCE*, as well as the previously described *NEGs*, into equal-sized training and test sets (*CCE-train, CCE-test, NEG-train, NEG-test*) and used them as improved reference sets to train our classifier and evaluate both data quality and analytical approaches in other data sets as well as screens withheld from our initial set of 72 cancer cell lines (Fig 6A).

## Improving analyses of shRNA pooled library screens

Our laboratory recently published a study of shRNA-driven synthetic lethality with several query knockout genes in an isogenic HCT116 colon cancer cell line background (Vizeacoumar *et al*, 2013). The HCT116 cell line is near diploid and thus does not suffer

from SCNA-driven biological artifacts. We trained our Bayesian classifier with CCE-train and NEG-train, applying a uniform prior (P(essential)/P(nonessential); see Materials and Methods) of 0.1, to yield a posterior log odds (LOD) of essentiality for each gene in each screen. Recall and precision were evaluated against CCE-test and NE-test, and an *F*-measure was calculated at a point on the curve where the LOD score crossed zero (Fig 6B). All six screens had *F*-measures > 0.8, adding confidence to analyses of essentiality and differential essentiality gleaned from these screens.

We applied the same analytical approach to the compendium of 102 pooled library shRNA screens from Project Achilles (Cheung *et al*, 2011), after filtering the reference sets for genes assayed by the 54k hairpin library used for those screens. Finding the point on the precision-recall curve where the LOD score crosses zero (Fig 6C, blue), we observe wide variability in the quality of the screens: only 65 of the 102 screens had an *F*-measure of 0.70 or greater. This variability in screen performance is consistent with that seen in the

Marcotte *et al* screens, and the lower overall *F*-measures may reflect bias from using different shRNA libraries rather than decreased performance (see below).

The reference sets can be used independently to evaluate any method of analyzing essentiality screens. For example, we used CCE-test and NEG-test to evaluate the published results of the ATARiS algorithm as applied to the Achilles data (Shao *et al*, 2012). After filtering the reference sets for genes with ATARiS solutions, genes from each screen were ranked by phenotype score, with the most negative score indicating strongest phenotype. Recall and precision were determined at a phenotype score of −1 (Fig 6C, red); generally, only a few hundred genes have stronger scores. ATARiS gave predictions for many cell lines that were worse than random, perhaps in part because it included all the Achilles data sets, including the lower quality ones. Filtering the input data for known performing screens could potentially improve scoring performance.

Using a different approach, Solimini *et al* (2013) analyzed the distribution of copy number changes of tumor suppressors ('STOP' genes) and essential genes ('GO' genes) across thousands of tumor-normal pairs. In the absence of a reference set of essential genes, the authors used two approaches to define GO genes: screening and theoretical. In the screening approach, the authors identified hairpins that dropped out in 5 of 9 library shRNA screens, yielding 1,127 genes with at least one hairpin. Though the single-hairpin approach is not widely accepted due to the frequency of off-target effects (Kaelin, 2012), the error rate is mitigated somewhat by requiring multiple observations across multiple screens. Evaluated against CCE-test and NEG-test, this set shows 49.5% recall and 16.9% FDR. The theoretical approach drew upon genes from selected core pathways in KEGG and yielded 545 genes with 54.1% recall and 1.7% FDR.

While the reference sets are broadly applicable to cancer functional genomics studies, the Bayesian approach used to classify essential genes can be readily extended to integrate other molecular data. We collected RNA-seq gene expression data on four pancreatic cancer cell lines withheld from our analysis of the COLT-cancer dataset and rank-ordered and binned (*n* = 500) genes by expression level. Within each bin, we plotted the mean expression (± s.d.) versus the fraction of genes in the CCE-train reference set (Fig 6D). We then used a linear fit to these data to calculate an expression-based informative prior for each gene, replacing the uniform prior used above in the calculation of LOD score (see Materials and Methods). Fig 6D shows the relationship between gene essentiality and expression in the CAPAN-2 cell line, while Fig 6E shows the relative performance of four analytical approaches evaluated against CCE-test and NEG-test. Applying the gene expression prior improved the performance of the screen in all cases (see Supplementary Fig S4 for the other 3 screens) over the LOD score with the uninformative prior and increased the margin of improvement over two current state of the art algorithms for library RNAi screens, GARP and RIGER. Thus, the combination of the reference set of core essentials and the Bayesian classifier offers a best-in-class method for analyzing such screens as well as a framework for integrating other molecular data to improve performance. Moreover, the core essentials offer a ready reference set against which to evaluate the relative performance of such screens.

## Evaluating CRISPR Negative Selection Screens

Gold-standard reference sets of essential and nonessential genes can be used to evaluate any large-scale assay of gene essentiality. Recently, the CRISPR system has been adapted to induce targeted genetic modification of human cells (Cong *et al*, 2013; Mali *et al*, 2013) and has been applied in genome-scale pooled library positive selection screens for specific pathway members and negative selection screens for essential genes. For example, Shalem *et al* (2013) recently published negative selection screens targeting 18,080 genes with ~65,000 guide sequences (gRNA) in two human cell lines, A375 melanoma cells and HUES62 embryonic stem cells. As with shRNA screens, a CRISPR gRNA targeting an essential gene will drop out of a population, resulting in a strong negative fold-change for that gRNA. As expected, the fold-change distributions of gRNA targeting training-set essential genes were left-shifted relative to the distributions of gRNA targeting nonessential genes (Fig 7A). We used these distributions to train our Bayesian classifier and evaluated our results against the withheld test sets. Fig 7B shows the improvement that the Bayesian classifier offers over the approach used in the original study. It also highlights the poor performance of the HUES62 screen, which explains the sparse overlap between the two screens reported in the original study.

Concurrently, Wang *et al* (2013) reported negative selection screens targeting 7,114 genes in two human cell lines, including the near-haploid KBM7 cell line. The performance curves of these screens, measured against CCE-test and NEG-test and shown in Fig 7C, are impressive but likely underestimate the actual error rates of these screens as core essential ribosomal genes are overrepresented and the nonessential reference set is severely underrepresented among target genes (~6-fold depletion relative to the Shalem *et al* library).

Taken together, these analyses offer two key insights into the differences between CRISPR and RNAi screens. The Bayes Factor analysis of the Shalem *et al* screen classifies 805 targets as essential at zero FDR. These 805 genes represent 47% recall of the reference essential set; extrapolation suggests there may be well over 1,600 essential genes in this cell line. As this is more than double the number of high-confidence essentials detected in most shRNA screens, and 50% more than the total number of essentials suggested by the cumulative analysis of RNAi screens, it suggests that CRISPR screens may have substantially greater sensitivity than pooled library shRNA screens.

We explored this finding by comparing both CRISPR and shRNA results to sample-matched gene expression data. In the Bayes Factor analysis of the four withheld pancreatic cancer cell line screens described above (without any expression prior to prevent circularity), the screens have an average of 664 genes with BF > 5 (range 584–740). We therefore defined the top 664 genes from each screen as hits. We then quantile normalized the corresponding gene expression values (rendering the distributions identical), rank-ordered each cell line's genes by expression level, and binned genes into groups of 500. For each screen, the mean expression level of genes in the bin was plotted against the fraction of genes in each bin that are classified as essential. Fig. 7D (red) shows the relationship between gene expression level and essentiality for the four shRNA screens. Genes with trace or zero expression (left edge of plot) cannot be essential, and hits in

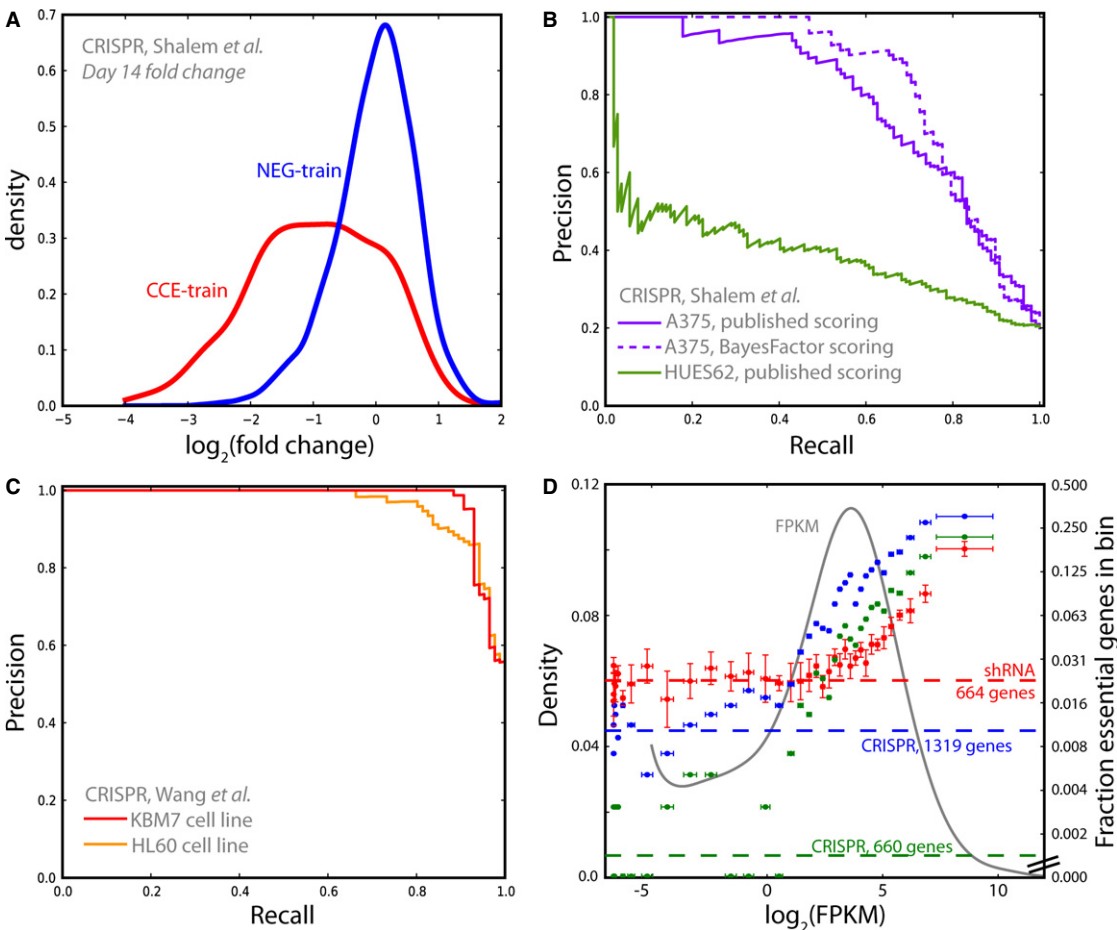

**Figure 7. Evaluating CRISPR negative selection screens.**

A The fold-change distributions of gRNA targeting reference essential and nonessential genes in Shalem *et al* (2013) are similar to those shown by shRNA hairpins (see Fig 1) and enable the application of the Bayes Factor approach.

B Published results from Shalem *et al* (2013), evaluated against CCE-test and NE-test. Dashed line shows that Bayes Factor approach more accurately captures essential genes in the A375 screen, the only screen for which raw data is available.

C Whole-screen results from Wang *et al* (Wang *et al*, 2013), evaluated against the same sets. NE-test genes are underrepresented in the Wang *et al* gRNA library, which gives the appearance of an artificial boost in precision when compared to the Shalem *et al* (2013) results.

D Comparing shRNA to CRISPR. Genes are rank-ordered by expression (gray curve, left axis) and binned. For four shRNA screens in pancreatic cancer cell lines withheld from the original analysis (red), the fraction of essential genes (by BF, no prior) in each bin ($\pm$ s.d., right axis) is plotted against the mean expression of all genes in the bin. Genes with trace expression ($\log_2$(FPKM) $< -2$) are not essential and can therefore estimate background error rate (dashed line). Comparing CRISPR results demonstrates that, for the one dataset available, CRISPR can yield a similar number of essential genes at ~10-fold lower FPR (green, BF $\geq$ 20, 660 genes), or double the number of essential genes at similar error rates (blue, BF $\geq$ 10, 1,319 genes).

this group are almost certainly false positives. The fraction of genes with expression $< -2$ that are classified as hits (Fig 7D, red dashed line, right axis) therefore estimates the screen's background error rate.

We compared this to the CRISPR Bayes Factor results described above. At a BF $\geq$ 20, we identify 660 essential genes, roughly the same number as the average of the shRNA screens. Plotting these genes as described above (Fig 7D, green), we observe a background error rate > 10-fold lower than that of the shRNA screens. Relaxing the threshold to BF $\geq$ 10 (Fig 7D, blue), we find 1,319 essential genes with a false positive rate comparable to, though still lower than, that of the shRNA screens. Though this is preliminary analysis of a single CRISPR screen measured against gene expression data from a different study, to a first approximation, the CRISPR technology shows a tenfold lower off-target rate at the same

coverage as shRNA, or double the coverage at a comparable error rate. Moreover, CRISPR appears to show increased sensitivity at lower, but still biologically relevant, expression levels.

Though CRISPR appears to offer a more accurate assay of gene essentiality, the error rate increases markedly after the top ~1,500 hits. False discovery rates of genome-scale CRISPR screens are largely unexplored in the first-generation published screens, but our analysis indicates that nontrivial numbers of false positives are indeed present in these screens. It is currently unknown whether these false positives arise from the technical variability inherent in large-scale screens or from the biological activity of off-target gRNA sequences. The reference sets and analytical methods we describe here offer a framework for understanding the nature of these false positives and, in turn, for refining the design of CRISPR gRNA libraries and experimental protocols.

## Minimizing bias by integrating data sets

We derived a preliminary set of core essential genes by finding essential genes in a majority of high-performing shRNA screens in the COLT compendium from Marcotte *et al.* The genes contained in this set are highly enriched for core cellular processes—in particular, they encode subunits of the protein complexes involved in transcription, translation, and replication—and have a very low incidence of false positives. This set improves on the results in Marcotte *et al*, which described 297 'general essentials' (293 with current gene IDs). The intersection of this set with our 291 core essentials ($n = 199$) shows a high proportion of genes with constitutive, invariant expression (81%). Of the genes unique to Marcotte *et al* ($n = 94$), only 42% have constitutive, invariant expression, compared to 61% of those unique to this study ($n = 92$), indicating higher accuracy.

These screens were all performed using the same shRNA library, and cell lines from only three cancer tissues of origin were assayed, likely yielding a biased summary of essential genes with an unknown number of false negatives. To minimize this bias, we integrated our results with those derived from an identical analysis of the Project Achilles screens, which were conducted with a different pooled shRNA library. Taking the 65 Achilles screens with *F*-measure > 0.70 (Supplementary Fig S5A), we identified the Bayes Factor threshold at which the average screen FDR was similar to the average screen FDR of the 46 COLT screens used above. At BF > 5, corresponding to an average screen FDR of 16%, we identified 345 genes that were essential in at least 33 of the 65 Achilles screens (Supplementary Fig S5B), of which 247 showed constitutive, invariant expression (Supplementary Fig S5C). Of these, 104 are the same

as those in the 217-gene COLT-derived CCE set, for a final set of 360 core essential genes. Genes unique to either set show similar proportions of constitutively expressed genes (63–65%), suggesting similar accuracy. The union of the two analyses is ~50% larger than the result from either data set alone, suggesting a substantial false negative rate for individual data sets derived from a single shRNA library.

## The Daisy model of gene essentiality

The mouse knockout data highlight an important factor in the study of gene essentiality. The definition of essentiality is context-dependent: a mouse (or human) gene may reasonably be classified as essential if its complete loss of function results in a phenotype ranging from prenatal to juvenile lethality or even sterility, with the onus on the researcher to explicitly define the term. Cell line assays of gene essentiality necessarily sample only the genes required for the proliferation of that cell line in cultured conditions; genes which may be required for organismal health may not be expressed in a given cell line and thus will not be detectable. Nevertheless, there is a core set of ubiquitously expressed, ubiquitously essential genes that should be detectable in virtually any cell line screen. This gives rise to the 'daisy model' of gene essentiality (Fig 8A), where each petal represents a cell-line- or tissue-specific context in which a gene's activity might be required. Petals will overlap to varying degrees but all will share the core set of essential genes. The core essentials described here represent our effort to define this set of universally essential genes.

The link between gene essentiality and genetic predisposition to disease has long been a topic of active study. We took the set of mouse knockout essentials and divided them into core and

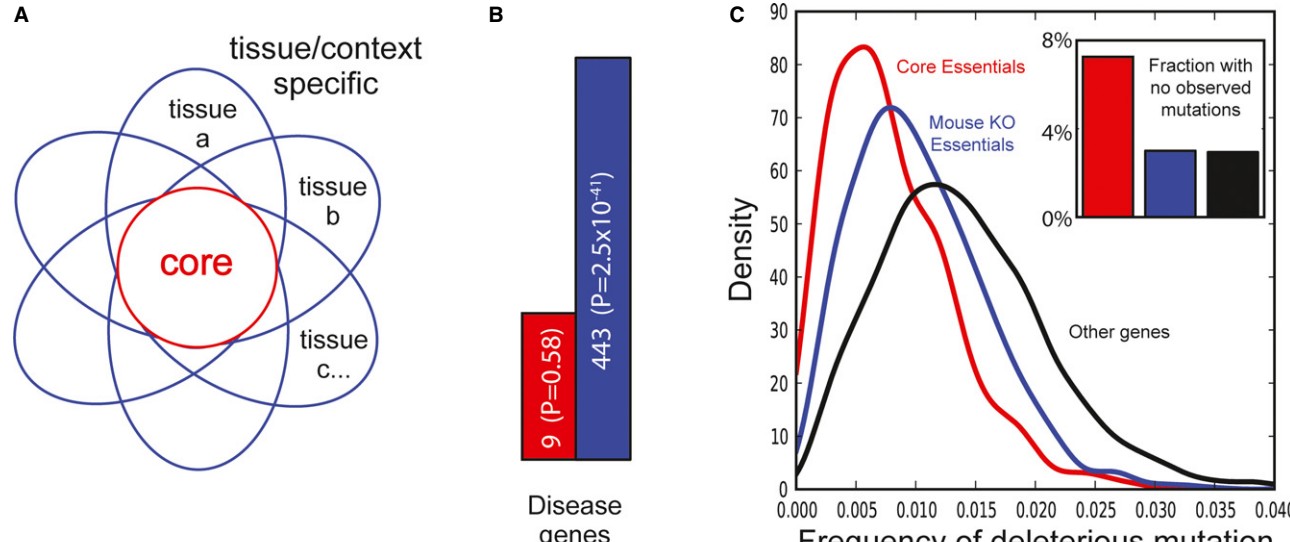

**Figure 8.  The Daisy model of gene essentiality.**

A  The Daisy model, where each petal represents a tissue or context in which a gene is essential. Petals overlap to varying degrees but all share a core set of essential housekeeping genes that should be detectable in any cell-based assay. Whole-organism studies will sample from the whole flower, not specific petals.

B  Human orthologs of mouse essential genes were divided into core and noncore ('peripheral') essentials. Peripheral essentials show strong enrichment for disease genes while core essentials do not.

C  Frequency of putative deleterious mutation by gene class, normalized for transcript length, derived from population exome studies (Tennessen *et al*, 2012). Inset, fraction of genes by class in which no variant was observed. Little variation is tolerated among core essentials, probably explaining the infrequency with which they are associated with disease.

peripheral essentials based on whether or not they were included in our integrated set of 360 core essential genes. Analyzing these sets for disease gene enrichment reveals that peripheral essentials are strongly enriched for disease genes ($P$ = 2.5e-41; Fig 8B), while core essentials show no enrichment beyond random expectation. This is consistent with previous findings (Lohmueller *et al*, 2008; Chavali *et al*, 2010; Dickerson *et al*, 2011), which gives rise to a model wherein core essential genes are less tolerant to genetic variation than peripheral essentials. The recent publication of large-scale human population genetic studies allows us to test these hypotheses. Fig 8C shows the rate of putative deleterious mutation observed in 2440 exomes (Tennessen *et al*, 2012), by gene class. Core essentials are much less likely to show deleterious variants than other essential or nonessential genes and are twice as likely to have no observed variant (Fig 8C, inset). This is further reflected when comparing our core essential genes to the human essential genes delineated by Liao and Zhang (Liao & Zhang, 2008). They find 120 human null mutations that give rise to juvenile lethality or sterility, which are organismal or peripheral essentials by our definitions, and as predicted show little overlap ($n$ = 3) with our core essentials.

## Discussion

In this study, we have generated a global set of essential genes in human cell lines based on experimental data. Drawn from genes that show consistent strong antiproliferative effects across a panel of pooled library shRNA screens in cancer cell lines, these essential genes are highly enriched for conserved protein complexes that carry out the fundamental work of the cell: transcription, translation, DNA replication, and protein degradation. Consistent with previous studies, these genes are more likely to be essential in mouse knockout studies and less likely to have a human paralog than other genes. We label these genes 'core essentials' as they are likely essential across all cell lines, tissue types, and developmental states.

We exploit the difference between core and peripheral, or context-specific, essentials in two ways. First, at the organismal level, we show that peripheral essentials, including human homologs of mouse essential genes, are more likely to be disease genes and demonstrate that core essentials show lower incidence of putative deleterious mutation in a normal human population. This finding explains a longstanding observation that human disease genes are enriched for whole-organism essentials but tend not to be housekeeping genes. That is, hypomorphic alleles of peripheral essentials cause a partial loss of fitness (i.e. disease), but hypomorphic alleles of core essentials are fatal. Cumulative analysis of RNAi screens suggests a total population of ~1,000 human cell line essential genes, while preliminary analysis of genome-scale CRISPR screens suggests roughly double this number, perhaps reflecting reduced sensitivity of RNAi methods against lower-expression genes.

Second, we derive the 'daisy model' of gene essentiality from the difference between core and context-specific cell line essentials, wherein each petal represents the set of essential genes in one cell line, tissue, or genomic context. Petals will overlap to varying degrees, but all contexts share the common core essentials. While the focus of essentiality studies in cancer cell lines is to find context-specific essentials that can provide highly specific therapeutic

targets, the degree to which a screen recapitulates the shared core essentials is a critical measure of its accuracy.

We used the core essentials, in conjunction with a set of putative nonessentials derived from the Illumina BodyMap and ENCODE studies of gene expression in human tissues and cell lines, as gold-standard reference sets to train a Bayesian predictor of gene essentiality in pooled library shRNA screens and to test our algorithm as well as several previously published algorithms and data sets. Our algorithm substantially outperforms other methods on the data sets we tested, particularly when coupled with sample-matched gene expression data. We also demonstrate that our method is applicable to other pooled library negative selection screens using CRISPR genome-editing technology and look forward to the onslaught of genome-scale screens that will emerge using this technology.

Our analyses reveal that copy number amplification in cancer cell lines can substantially decrease a core essential gene's sensitivity to RNAi perturbation. This is most likely driven by the encoded protein's membership in a protein complex: genomic amplification leads to over-expression and protein abundance beyond the stoichiometric requirements for complex function. Interestingly, the converse is also true: hemizygosity increases sensitivity. A recent study found that partial loss of some genes in tumors resulted in increased vulnerability to perturbations of those genes—the so-called CYCLOPS genes (Nijhawan *et al*, 2012). As CYCLOPS genes are enriched subunits of core essential complexes, our findings may extend the CYCLOPS concept to all core essential complexes. That is, copy number losses among essential subunits may render cancer cells more susceptible to pharmacological compounds targeting these complexes. This concept may apply to expression-sensitive enzymes as well.

Broadly speaking, the reference sets of cell line essential and nonessential genes we provide represent a useful yardstick against which cancer functional genomics studies can be measured. Lack of such suitable yardsticks has contributed to critical errors in the field, including high profile reports of synthetic lethal interactions with common oncogenes (Scholl *et al*, 2009) that were later disproven (Babij *et al*, 2011; Luo *et al*, 2012; Weiwer *et al*, 2012) (and also do not appear in our data), and has led to a reassessment of shRNA methodologies (Kaelin, 2012). Such gold-standard reference sets will become increasingly important as the CRISPR genome-scale genetic perturbation technology matures (Cong *et al*, 2013; Mali *et al*, 2013). Our analysis of available data indicates that CRISPR screens can be more sensitive than RNAi methods in detecting essential genes, but that CRISPR library screening is also subject to a non-trivial false discovery rate—a finding that is largely ignored in the current literature. Progressively improving performance against an established set of benchmarks is the best way to validate such new technologies and their accompanying analytical methods, to ensure their widespread adoption, and to unlock the biological discovery that their application enables.

## Materials and Methods

### Software

A collection of python scripts and sample data is available as a supplementary archive file (Supplementary Software Package). The

archive contains all the scripts, data, and reference sets necessary to calculate Bayes Factors for one cell line.

## Using Matrix Decomposition to find a seed set of putative essentials

The 72 pooled library shRNA screens were divided into three sets: group one ($n = 34$), group two ($n = 34$), and withheld ($n = 4$; see sample key in Supplementary Dataset S1). For screens in group one, all repeats from all timepoints were combined into a fold-change matrix of ~78,000 hairpins by ~200 arrays. Singular value decomposition was performed on the matrix; the top singular value was found to explain > 40% of the total variance of the matrix (see Supplementary Fig S1). Hairpins with strong positive projections onto the first left singular vector (U1) showed strong negative fold-change across most of the 34 samples in the group one matrix.

We used a statistical filter to find genes enriched for hairpins with strong U1 projections. For each gene, hairpins were rank-ordered by U1 projection, and the median projection $p$ among hairpins targeting the gene was determined. Then, the enrichment $P$-value was calculated by the hypergeometric test:

$$P(\text{enrichment}) = \text{hypergeometric}(X >= x | n, m, N).$$

where $x$ is the rank of the median hairpin for the gene; $n$ is the number of hairpins targeting the gene; $m$ is the total number of hairpins in the population with U1 projection $>= P$; and $N$ is the total number of hairpins in the experiment.

Adjusted $P$-values were calculated by the method of Benjamini & Hochberg, and genes with adjusted $P$-value < 0.25 were selected as putative seed essentials. This list was further filtered for genes with constitutive, invariant gene expression across two sets of RNA-seq data, the ENCODE set of 17 human cell lines, and the Illumina BodyMap set of 16 healthy human tissues (see RNA-seq analysis, below).

## RNA-seq analysis

We used Tophat v1.4.1 to align RNA-seq reads to the hg19 human transcriptome defined in the Gencode v14 GTF file, using default Tophat parameters. We used Cufflinks in quantitation-only mode with the same GTF file to generate FPKM values for each gene. FPKM values were filtered for protein-coding genes (as defined by HGNC, www.genenames.org) and log-transformed (adding 0.01 as a pseudocount). The mean log(FPKM) of technical or biological repeats was used, where applicable (e.g. biological repeats in ENCODE and technical repeats at $2 \times 50$ and $1 \times 75$ read type for BodyMap).

For ENCODE (GEO accession GSE30567) and BodyMap (EBI accession E-MTAB-513), constitutive, invariant genes were defined as genes with mean expression in each data set > 0 and standard deviation < mean standard deviation across all protein-coding genes. Genes must be constitutive and invariant in both data sets. The reference set of putative nonessential genes is defined as protein-coding genes with FPKM < 0.1 in 15 of 16 BodyMap tissues *and* FPKM < 0.1 in 16 of 17 ENCODE cell lines. The set is filtered for genes that are assayed by the pooled shRNA library.

## Calculating the Bayes factor

Seed essentials from SVD of group one and nonessentials from gene expression were divided into equal-sized sets for training and testing, and used to train and evaluate the classifier for each cell line in group two (and vice versa). Each cell line was assayed at two timepoints. For each timepoint, a density function of the fold-changes of all hairpins targeting essential genes in the training set was estimated by Gaussian kernel density estimation using the scipy.stats.gaussian_kde function in Python. The process was repeated for nonessential genes. Then, for each gene, the Bayes Factor is calculated as follows:

$$\text{BF} = \frac{\Pr(\text{data} \,|\, \text{essential})}{\Pr(\text{data} \,|\, \text{nonessential})} = \prod_{i,j} \frac{\Pr(\text{fc}_{i,j} \,|\, \text{essential})}{\Pr(\text{fc}_{i,j} \,|\, \text{nonessential})}$$

across hairpin observations $i$ and timepoints $j$, where $\Pr(x)$ is the density function.

Log-transforming the equation yields:

$$\log(\text{BF}) = \sum_{i,j} (\log(\Pr(\text{fc}_{i,j} \,|\, \text{essential})) - \log(\Pr(\text{fc}_{i,j} \,|\, \text{nonessential})))$$

For a typical gene with 5 cognate hairpins assayed with three biological repeats, the log(BF) is the sum of 15 values at each of two timepoints.

Contributions to the BF score can be dominated by high fold-change hairpins, where the Pr(data | nonessential) term is very small. To prevent these outliers from dominating the final BF, we empirically truncate $\log_2$ fold-changes at $-4$ and $+0.5$. This keeps individual hairpin contributions to the BF within a reasonable dynamic range, and greater absolute fold-changes do not provide substantially greater evidence for or against essentiality.

## Using priors to calculate posterior log odds

A Bayes Factor can be extended to a posterior odds ratio by multiplying by an appropriate ratio of priors:

$$\text{OR} = \frac{\Pr(\text{data} \,|\, \text{essential})}{\Pr(\text{data} \,|\, \text{nonessential})} \times \frac{\Pr(\text{essential})}{\Pr(\text{nonessential})}$$

$$\log(\text{OR}) = \log(\Pr(\text{data} \,|\, \text{essential})) - \log(\Pr(\text{data} \,|\, \text{nonessential})) + \log(\text{prior ratio})$$

Where indicated in the main text that a posterior log odds ratio (LOD score) was calculated (e.g. the withheld group, the HCT116 screens, and the Achilles screens), a uniform prior ratio of 0.1 was applied (by adding $\log_2(\text{prior}) = -3.32$ to each logBF), representing a background expectation that ~10% of assayed genes are essential.

For samples in the withheld group, we also calculated a specific prior for each gene based on its expression level. We generated $\log_2(\text{FPKM})$ values for all protein-coding genes as described above. Genes were rank-ordered by expression level and binned ($n = 500$). For each bin, we calculated the mean expression level of genes in the bin and the $\log_2$ of the fraction of genes in the CCE-train reference set, adding a pseudocount of 0.001 to prevent infinities (see Fig 6D). A linear fit was applied

to bins with mean expression > 1. This linear fit was used to calculate an expression-based prior, with the log₂(fraction essential genes in bin) approximating the log-prior described above.

### Evaluating precision and recall for each screen

For each screen, the applicable reference sets were divided into equal-sized training and testing sets. Training sets were used to estimate the density functions of essential and nonessential hairpin fold-changes, as described above, and (where applicable) to calculate the expression-based prior. Withheld testing sets were used to evaluate the performance of each screen.

Genes from each evaluated screen were rank-ordered by Bayes Factor or LOD score, which ever was applicable. Then, for each gene, the cumulative precision and recall were calculated as Recall = TP/(TP + FN) and Precision = TP/(TP + FP), where TP = true positives, the number of genes in the essentials test set with BF/LOD score greater than the current gene; (TP + FN) = the total number of essentials in the test set; and FP = false positives, the number of genes in the nonessentials test set with BF/LOD score greater than the current gene.

The *F*-measure was calculated as a single, global metric for screen quality. The *F*-measure is the harmonic mean of precision and recall calculated at a specified BF/LOD (typically 0):

$$F = 2 \frac{(\text{precision} \times \text{recall})}{(\text{precision} + \text{recall})}$$

To evaluate the ATARiS results, we used phenotype scores from Achilles_102lines_gene_solutions.gct (downloaded from http://www.broadinstitute.org/achilles/). For each screen, genes were rank-ordered by phenotype score and precision and recall were calculated as above, using the CCE-test and NE-test reference sets. *F*-measure was calculated at phenotype score = −1.

### Absolute copy number

SNP analysis was performed at the University Health Network Microarray Center (Toronto, ON, CA) using Illumina (Illumina, San Diego, CA) HumanOmni1 BeadChip according to manufacturer's instructions. Normalized LogR ratio (LRR) and B allele frequency (BAF) signals for each probe were exported from the Illumina BeadStudio utility. Export files were then processed with the Genome Alteration Print (GAP) algorithm (Popova *et al*, 2009). Projections of LRR and BAF profiles were created, and pattern recognition was performed for each samples. Parameters were set as followed: germHomozyg.mBAF.thr > 0.97 and p_BAF = 0 (no normal contamination). Each pattern was visually inspected and corrected when the grid was off the segment center clusters. Output files produced by GAP were processed in order to obtain segments defined by copy number change only. Briefly, adjacent segments with identical absolute copy number were merged, and the LRR values were averaged. Gene level absolute copy number and LRR were obtained using the CNTools package.

**Supplementary information** for this article is available online: http://msb.embopress.org

## Acknowledgements

The authors would like to thank all the members of the Moffat and Rottapel labs for helpful discussions. This work was supported by the Ontario Ministry of Research and Innovation's GL2 Program and the Selective Therapies Program at the Ontario Institute for Cancer Research. JM is a Tier II Canada Research Chair in Functional Genetics and a Research Fellow at the Canadian Institute for Advanced Research.

## Author contributions

TH and JM conceived of the ideas and wrote the manuscript with input from KRB, FS, and RR.

## Conflict of interest

The authors declare that they have no conflict of interest.

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
