## [Review Process File · Molecular Systems Biology]

Measuring error rates in genomic perturbation screens: gold standards for human functional genomics

Traver Hart, Kevin R. Brown, Fabrice Sircoulomb, Robert Rottapel, Jason Moffat

Corresponding author: Jason Moffat, University of Toronto

Review timeline:	Submission date:	20 February 2014
	Editorial Decision:	14 March 2014
	Revision received:	10 April 2014
	Accepted:	24 April 2014

Editor: Maria Polychronidou

Transaction Report:

1st Editorial Decision

14 March 2014

Thank you again for submitting your work to Molecular Systems Biology. We have now heard back from the four referees who agreed to evaluate your manuscript. As you will see from the reports below, the referees acknowledge that the presented approach is potentially useful for functional genomics analyses. However, they raise a series of concerns, which should be carefully addressed in a revision of the manuscript.

Without repeating all the points listed below, among the more fundamental issues are the following:

- The reviewers refer to the need to address the potential bias associated with the shRNA libraries and the cell lines used in the screen.
- Including a more direct comparison of RNAi and CRISPR screens would significantly enhance the study.

Moreover, reviewers #1 and #3 point out that providing a software/code that can be used for implementing the presented approach would increase the impact of this work.

If you feel you can satisfactorily deal with these points and those listed by the referees, you may wish to submit a revised version of your manuscript. Please attach a covering letter giving details of the way in which you have handled each of the points raised by the referees.

Reviewer #1:

The paper describes statistically sound methodology to analyze shRNA screens. The idea is to define sufficiently large sets of positive and negative 'ground truth' (genes whose knock-down definitely has or has not a viability effect) and use these to calibrate detection parameters, estimate FPR, FNR etc.

The definition of such sets is made possible by analysing a large dataset previously published by the lab. A Bayes factor approach to hit scoring is introduced.

The paper addresses a fundamental, conceptually simple, but practically still challenging and essentially unsolved problem. Its approach is technically sound, the presentation is thoughtful, I believe that this is an important piece of work of great interest to the functional genomics community, both for researchers immediately interested in applications of shRNA screens, but also for researchers working in method design for screening technologies more generally.

I would consider it essential for the impact of this paper that the software be made available that implements the proposed approach; as well as a 'literate programming' vignette that demonstrates the computations on the dataset.

Beside, I only have a few technical remarks, specifically, the presentation of the part of the methods that is about statistics could be edited by a mathematical statistician to ensure its accuracy.

p.22 The summation for $\log(\text{BF})$ involves many individually large terms (since the logarithm of a small positive value can be very large in absolute terms), would the method be more robust (and more performant) if a trimmed summation (as in the trimmed mean) were used to estimate the BF?

p.13 "Leveraging gold-standard reference sets to improve analyses of CRISPR and shRNA screens." It took me some time to understand the relationship between this section and the definition of the reference set of 148 on p.7 and Fig.1. Perhaps this could be better structured, and explained in a schematic figure.

p.10 "Assuming that the first 12 screens..." - the wording here is a bit opaque, as the assumption is obviously not true - but helps to arrive at an upper boundary for the FDR. I suggest that this could be worded more clearly for non-statisticians.

p.11/12 "we found that AGO2 was the top ranked correlation" - a gene is not a correlation, rephrase?

p.19 While I am no expert, I am not sure that the KRAS-STK33 story would have played out differently if a reference set of essential/non-essential genes had been available. This claim could be toned down.

p.21 I did not understand the terminology "median hairpin projection", this could be rephrased for clarification.

p.21 "Q-values were calculated from P-values by the method of Benjamini & Hochberg" - The term 'q-value' is often used for results of Storey's method, whereas the authors of the BH method seem to prefer the term 'adjusted p-values'.

p.21 How was 0.01 chosen as a pseudocount? Does this not introduce huge granularity at the lower end of the dynamic range. How sensitively do the downstream results depend on this parameter choice?

p.22 "Seed essentials from SVD of group one and nonessentials from gene expression were divided into equal-sized sets for training and testing" -- it seems needlessly inefficient to waste half of the data for testing (and not to use them for training). A more standard machine learning approach is to estimate classifier performance via cross-validation and then determine the actual classifier on all data (see e.g. Hastie, Tibshirani, Friedman's book).

p.22 "For each timepoint, an empirical distribution of the fold-changes of all hairpins targeting essential genes in the training set was calculated using the `scipy.stats.gaussian_kde` function in Python." - the empirical distribution is just that, empirical, a sum of point masses, it is not smoothed out by a kernel density estimator. If a smooth density estimate is needed, then please state the estimator explicitly (in mathematical terms).

(minor:) Italic font in formulae should be reserved for symbols. Plain words should be written as

such, to avoid confusion of 'fc' with 'f*c' or 'essential' with e * s * s * e * n * t * i * a * l.

Reviewer #2:

Summary

The manuscript by Hart et al. proposes a framework for evaluating the quality of genome-scale lethality screens by assembling reference sets of essential as well as non-essential genes and providing Bayesian classifier of gene essentiality. Using this framework, the authors evaluated several published RNAi screens and indicated that this framework outperforms current methods. In addition, the authors also applied this framework to datasets generated with CRISPR technology and compared the sensitivity as well as specificity of RNAi screens compared to CRISPR screens. The work is of interest, however, many issues need to be addressed.

General remarks

Essential genes are the genes that are critical for survival. Understand essentiality is important to characterize biological systems and to devise strategies to treat cancer. Essential genes have been well characterized for many prokaryotic organisms but not for eukaryotic species. *Saccharomyces cerevisiae* is the only eukaryotic species, for which systematic essentiality screens have been carried out, revealing that 15-20% of all genes are essential. Human gene essentiality has been studied in cell lines by loss-of-function RNAi screens and, more recently, CRISPR screens. But the major hurdle for these technologies is the lack of specificity. Off-target activities complicate the interpretation of screen results and currently there is not efficient way of evaluating different studies. The work by Hart et al. will help data analysis of lethality screens by providing gold standard reference gene sets of essential/non-essential genes. As Hart et al. said in the manuscript, "a useful yardstick against which cancer functional genomics studies can be measured". Currently, there are only a couple hundred essential genes annotated in the human genome. This study will be a great resource for essential human genes as well as a great step forward for evaluating lethality type screens using RNAi and CRISPRs.

Major issues to be addressed by the authors

1. All the essential gene sets assembled were based on pooled shRNA screens using the TRC library. Since there maybe certain bias and issues associated with each library, it would be informative to see the analysis done with other RNAi libraries.
2. OTEs have been linked to the mechanism of action of miRNAs, in which the 'seed region', a short sequence of bases 2-8 on the 5' end of the RNAi duplex, is complementary to the 3' untranslated regions (UTRs) of multiple mRNAs, causing degradation of their associated transcripts. Usually this type of OTEs is phenotype specific. Sigoillot et al (A bioinformatics method identifies prominent off-targeted transcripts in RNAi screens. 2012, Nature method) showed that many top hits of the spindle assembly checkpoint screen analyzed have seed region match to the 3'UTR of MAD2, therefore are OTEs. Thus, the authors should go back to the sequences of RNAi reagents of the essential genes identified and perform a seed region analysis.
3. All the essential genes identified in this study were based on the RNAi data of cancer cell lines. The cellular background is quite different between cancer cell lines and normal cell lines, thus the study should be performed on normal cell line data as well.
4. Negative gene sets assembled were based only with genes that are not expressed. This set is potentially biased for genes that are not properly annotated or not detected due to technical issues during experiments. Thus, it is important to include genes that are expressed but do not have lethality phenotypes.
5. There are about 120 human essential genes annotated from literature at DEG (Database of Essential Gene, <http://www.essentialgene.org/>). There is not much overlap of DEG gene list with the essential genes sets assembled in this study. It would be great for the authors to provide the explanation of the lack of overlap and why they were not included.
6. Authors are reporting an interesting observation that AGO2 was the top ranked correlation when measuring the correlation between gene expression and screen F-measure, suggesting that AGO2 mRNA level may explain why some screens perform better than others. It would be great if the authors could further confirm this observation by providing the data of AGO2 expression by either western blot or qPCR in several cell lines.
7. The authors observed a negative correlation between copy number and essentiality. They argued that "increased copy number yields protein levels in excess of stoichiometric requirements for protein complex function", which may explain false negatives in essential genes in some screens.

Copy number does not always reflect the expression level of a gene, so it would be more convincing to show the expression data of these genes.

Minor issues to be addressed by the authors

1. When the seed reference set of essential genes was assembled, the authors selected the expressed genes based on NGS data from 17 cell lines and 16 BodyMap tissues. The expression threshold used was RPKM = 1. This threshold seems low and needs to be justified.
2. The authors compared the human essential genes with the essential genes from mouse KO studies. It would be interesting to compare human essential genes with yeast essential genes as well.
3. Typos: On page 15, paragraph 1, figure 6a should be figure 7a. On page 15, paragraph 2, figure 6b should be figure 7b.

Reviewer #3:

The manuscript by Hart et al. describes a useful method for the analysis of genetic screens both RNA-i based and CRISPR-based.

The authors derive what they call "gold-standard reference sets of essential and non-essential genes", mainly based on the analysis of RNAi screens across multiple cell lines, described previously by the same group in Marcotte et al., 2012. The analysis provides a useful set of Core Essential genes, which are predicted to be essential in most (all) human cells and a broader set of Total Essential, comprising Essential genes which are necessary only in a subset of cell lines. They also identify a set of putative Non-Essential genes.

These gene sets are then used by the authors to re-evaluate every single cell line as an individual screen in the dataset described in Marcotte et al. Interestingly --and somewhat expectedly-- they found that two main sources of variability and of high rate of false positives in the screens are low AGO2 expression and high copy number levels (amplifications) for single genes. These are important aspects when screens are performed in non-isogenic highly aneuploid cell lines.

Below are 5 minor comments and suggestions for potential improvements of the paper. They are in order of appearance in the paper but the most important one is point 3

1) The two sets of Essential and Non Essential genes are at least partially biased for the RNAi library and types of cell lines used in the screens, whose analysis is utilized to derive the gene sets themselves (Marcotte et al paper). Even though the screen-dependent bias should be buffered by the simultaneous analysis of multiple cell lines, it is not possible to exclude the presence of such a bias. Thus, since the authors define these two gene sets "gold-standard reference sets of essential and non-essential genes", they should at least add a note within the text stating the fact that these two lists are subject to potential weakness of the RNAi screen methodology itself, so most likely are not comprehensive and are necessarily dependent on the screen used to derive them. Thus, even though very useful, they cannot represent an absolute "gold standard" but a very good list approaching a gold standard..

This bias may come into play when for example the authors evaluate the performance of the screens described in Cheung et al, 2011 using these gene sets.

A better evaluation of these screens would be using in silico gene sets or after generation of another Essential and Non essential gene sets based on these screens to be evaluated (i.e repeating the analysis that the authors perform on the screens described by Marcotte et al., 2012 in the first part of the text).

Related to this is the fact that in theory the "gold standard" sets of Essential and Non Essential genes could be improved by including in the analysis all the RNAi screens (Marcotte et al , Cheung et al , Solimini et al.) and potentially CRISPR screens described in the text.

2) The authors develop a new method for evaluating the screens described in Marcotte et al., 2012, using the newly derived Essential and Non Essential gene sets.

They should compare this method with others standard methods commonly used and previously described.

For example another method of evaluation of a screen is by looking "simply" at the enrichment of an in silico list of Essential genes among the genes that are dropping out in a screen, similar to what

was described in Marcotte et al., 2012. It would be good if the authors could evaluate each screen (each cell line) in the same screen dataset of Marcotte et al using this method and compare it with the more sophisticated method that they describe. It would be good to know whether the 48 cell lines performing well using the newly developed method would have been identified also with simpler methods or not.

3) The third issue concerns the evaluation of the CRISPR screen. The authors say that the screen published by Shalem et al. has a N of genes classified as Essential (by their BF methods) that is higher than the cumulative analysis of all the cell lines screened by RNAi. From this analysis the authors suggest that CRISPR screens have greater sensitivity and report this in the abstract as well. The analysis of the CRISPR screen in comparison with RNAi is an important point but should be improved. It would be important if the authors compared RNAi and CRISPR screens done using the same cell line, the melanoma cell line A375, for which the Shalem et al. paper and a previous paper describe both types of screening methods.

This exact type of comparison was not done in the Shalem et al. paper and given the expansion of CRISPR technology would be of great interest for many researchers.

For this type of analysis it would be good to use both the "gold standard" sets derived by the authors and also a theoretical list of Essential genes.

4) It would be great if the authors could provide an automatic tool online for the evaluation of a genetic screen.

5) A curiosity: Since the authors describe a correlation between lack of efficacy of shRNA in the detection of an Essential gene and the copy number level of that gene in the cell, I was wondering whether this effect may be even more prominent using the CRISPR technology, since it is based on gene inactivation at the genomic level.

Reviewer #4:

In this manuscript the authors describe the selection of sets of essential and non-essential genes based on the results of a large number of genome scale shRNA screens in a large panel of cell lines. They propose the use of these essential genes as a standard for quality assessment of functional genomic screens. Selection of a seed set of essential genes is based on singular value decomposition of a large matrix comprising shRNA screen viability data. From this, a median hairpin projection was determined, followed by calculation of an enrichment P-value by a hypergeometric test and corrected Q-values. This initial essential gene set was further filtered using expression analysis. The selected essential and non-essential genes were used to calculate a Bayes Factor score for each gene in each cell line and subsequently an F-measure for screen performance. Using the group of high performing screens, the authors selected 291 genes scoring as essential in more than half of the cell lines, and considered this a core essential list. With this set, an estimation of false discovery rates is made and used to extend the list to 823 genes with an estimated FDR of 6-11%. These 823 genes are enriched for functional protein complexes involved in core biological processes. This set of genes is further used to indicate a negative correlation between gene copy number and essentiality. Finally, the authors define a set of 217 constitutive core essential genes that are used to evaluate other large scale screens, including shRNA and CRISPR based screens.

In conclusion, this manuscript describes the generation of a reference set of cell line essential and non essential genes (based on shRNA screening results) that can be used for quality assessment of functional genomic screens with the goal to validate new technologies and to ensure the appropriate use, interpretation and conclusion of such efforts.

The work described in this manuscript is of high interest to the functional genomic screening community and addresses an important issue of quality standards for large scale screening projects. However, to ensure broad-readership, the analytical methods should be more clearly described and where possible illustrated with an explanatory figure. It would for example be helpful to present a figure that shows on which parts of the large screening matrix the different stages of the analytical procedure were performed and to what extent the essential gene sets identified with the different methods correlate (overlap) with each other.

The selection of essential genes is solely based on the results of shRNA screens. Although there is ample attention to possible off-target effects, there is no discussion of the limitation of the technology with respect to insufficient knock-down and consequential false negative rates. This point should be discussed, also in the context of the other gene editing technologies. Furthermore, to further substantiate the relevance of the identified essential genes, the authors show that many of these genes belong to essential complexes. One could imagine that this "prior knowledge" is also incorporated in the selection of core essential genes. This would also allow for an estimation of false negative rates for other functional genomic technologies.

On page 12, the authors discuss a possible correlation between AGO2 expression levels and screen F-measure. This observation is intriguing but rather weak and potentially biased by cell type of origin. To strengthen this conclusion and drastically improve the utility of the presented work to the screening community, this could be tested experimentally by comparing F-measures of screens done in the same cell line, before and after over-expression of AGO2.

At the bottom of page 12, the authors suggest that the use of CRISPR technologies could test the role of copy number variation in false negative rates for shRNA screens. However, this technology may also suffer from the same problem where editing of multiple alleles can be rate limiting for this technology.

On page 19, the authors address a potential explanation for the effect of copy number variation and essential gene sensitivity to RNAi perturbation. They state that this can be explained by the fact that these proteins are part of a protein complex. However, one could argue that also non-complexed proteins with e.g. an enzymatic function can increase in expression above the required level of activity thereby reducing the effects of perturbation by RNAi. The conclusion that the findings described in this paper "extend the CYCLOPS concept to all core essential complexes" is not substantiated by the data provided in this manuscript and should be adjusted accordingly.

Finally, the title of this manuscript states that a gold standard has been established with this work. I would argue that this is an exaggeration as it builds on the results of shRNA screens with their specific limitations. The 'golden standard' list presented here will likely change if for example performed on a large set of CRISPR screens. It would be more appropriate to change the title and all references of a golden standard to for example ": a tool for standardizing human functional genomic screens".

Minor point:

The legend of Figure S1C mentions projections on the left singular vector where the annotation in the figure states right singular projection. This inconsistency should be corrected.

1st Revision - authors' response

10 April 2014

Response to Reviewers

Reviewer #1:

The paper describes statistically sound methodology to analyze shRNA screens. The idea is to define sufficiently large sets of positive and negative 'ground truth' (genes whose knock-down definitely has or has not a viability effect) and use these to calibrate detection parameters, estimate FPR, FNR etc.

The definition of such sets is made possible by analyzing a large dataset previously published by the lab. A Bayes factor approach to hit scoring is introduced.

The paper addresses a fundamental, conceptually simple, but practically still challenging and essentially unsolved problem. Its approach is technically sound, the presentation is thoughtful, I believe that this is an important piece of work of great interest to the functional genomics community, both for researchers immediately interested in applications of shRNA screens, but also for researchers working in method design for screening technologies more generally.

I would consider it essential for the impact of this paper that the software be made available that implements the proposed approach; as well as a 'literate programming' vignette that demonstrates the computations on the dataset.

We agree with the Reviewer and have included an archive file containing all the scripts, reference sets, and raw data to calculate BFs for one of the cell lines described in the paper (SK-PC-1). This package can be used to re-calculate BFs from primary data; for example, at <http://dpsc.ccb.utoronto.ca/cancer/> or using other available datasets.

Beside, I only have a few technical remarks, specifically, the presentation of the part of the methods that is about statistics could be edited by a mathematical statistician to ensure its accuracy.

This is a good suggestion and, as a result, have had the modified Methods section proofread by a mathematician for accuracy.

p.22 The summation for $\log(\text{BF})$ involves many individually large terms (since the logarithm of a small positive value can be very large in absolute terms), would the method be more robust (and more performant) if a trimmed summation (as in the trimmed mean) were used to estimate the BF?

This is an excellent observation that we failed to completely address in the manuscript. We do, in fact, prevent individual hairpin observations from dominating the BF calculation. In particular, the term $\log(\text{Pr}(\text{data} \mid \text{nonessential}))$ can get very large in absolute terms where observed hairpin fold changes (for, e.g., highly essential genes) lie far outside the range of fold changes observed in the nonessential training set. We mitigate this by truncating negative fold change at -4 and positive fold change at 0.5. Fold changes of greater magnitude are not more informative, and the contribution of individual hairpin observations to the total BF is not dominated by outliers. Text describing this has been added to the methods section.

p.13 "Leveraging gold-standard reference sets to improve analyses of CRISPR and shRNA screens." It took me some time to understand the relationship between this section and the definition of the reference set of 148 on p.7 and Fig.1. Perhaps this could be better structured, and explained in a schematic figure.

In order to clarify this approach/analysis an explanatory schematic panel has been added to Figure 6 (new Figure 6a).

p.10 "Assuming that the first 12 screens..." - the wording here is a bit opaque, as the assumption is obviously not true - but helps to arrive at an upper boundary for the FDR. I suggest that this could be worded more clearly for non-statisticians.

We have modified the text as follows: "If we assume that the first 12 screens have achieved saturation—an obviously false assumption but a useful approximation for modeling—then all subsequent hits must be false positives. The second and third sets of 12 screens therefore model the frequency distribution of false positives and give an estimate of the expected number of false positives in each bin."

p.11/12 "we found that AGO2 was the top ranked correlation" - a gene is not a correlation, rephrase?

We have modified the text as follows: "we found that AGO2 expression *had* the top ranked correlation..."

p.19 While I am no expert, I am not sure that the KRAS-STK33 story would have played out differently if a reference set of essential/non-essential genes had been available. This claim could be toned down.

We have modified the text as follows: “Lack of such suitable yardsticks has contributed to critical errors in the field...”

p.21 I did not understand the terminology "median hairpin projection", this could be rephrased for clarification.

We have modified the text as follows: “For each gene, hairpins were rank-ordered by U1 projection, and the median projection p among hairpins targeting the gene was determined.”

p.21 "Q-values were calculated from P-values by the method of Benjamini & Hochberg" - The term 'q-value' is often used for results of Storey's method, whereas the authors of the BH method seem to prefer the term 'adjusted p-values'.

We thank the Reviewer for pointing this out and have changed the appropriate text to “adjusted P-values.”

p.21 How was 0.01 chosen as a pseudocount? Does this not introduce huge granularity at the lower end of the dynamic range. How sensitively do the downstream results depend on this parameter choice?

The pseudocount has virtually no effect on biologically relevant levels of gene expression, which is typically FPKM of 0.1 or above (at the minimum), as described previously [PMID 24215113]. Moreover, RNA-seq profiles shown in Figure S2a, S2b, newly ordered Figure 6, and newly ordered Figure 7 show that genes of interest are almost always expressed at FPKM ≥ 1 . The pseudocount is therefore $< 1\%$ of gene expression for most relevant genes.

p.22 "Seed essentials from SVD of group one and nonessentials from gene expression were divided into equal-sized sets for training and testing" -- it seems needlessly inefficient to waste half of the data for testing (and not to use them for training). A more standard machine learning approach is to estimate classifier performance via cross-validation and then determine the actual classifier on all data (see e.g. Hastie, Tibhsirani, Friedman's book).

In principle we agree, but the structure of our data lends itself to our approach. Taking ~ 100 seed essential genes and dividing into halves yields 50 genes each for training and testing. The training set is actually the hairpin data for those genes, representing on average 5 hairpins \times 2 timepoints \times 3 repeats or 30 observations per gene, for $\sim 1,500$ training data points. A traditional multi-fold cross-validation increases this already sizeable training data set at the cost of a much smaller test set, which substantially increases the granularity of evaluation of the withheld cross section.

p.22 "For each timepoint, an empirical distribution of the fold-changes of all hairpins targeting essential genes in the training set was calculated using the scipy.stats.gaussian_kde function in Python." - the empirical distribution is just that, empirical, a sum of point masses, it is not smoothed out by a kernel density estimator. If a smooth density estimate is needed, then please state the estimator explicitly (in mathematical terms).

We thank the reviewer for pointing out our incorrect language. We have modified the text as follows: “For each timepoint, a density function of the fold-changes of all hairpins targeting essential genes in the training set was estimated by Gaussian kernel density estimation using the `scipy.stats.gaussian_kde` function in Python.”

*(minor:) Italic font in formulae should be reserved for symbols. Plain words should be written as such, to avoid confusion of 'fc' with 'f*c' or 'essential' with $e * s * s * e * n * t * i * a * l$.*

We have modified the text accordingly.

Reviewer #2:

Summary

The manuscript by Hart et al. proposes a framework for evaluating the quality of genome-scale lethality screens by assembling reference sets of essential as well as non-essential genes and providing Bayesian classifier of gene essentiality. Using this framework, the authors evaluated several published RNAi screens and indicated that this framework outperforms current methods. In addition, the authors also applied this framework to datasets generated with CRISPR technology and compared the sensitivity as well as specificity of RNAi screens compared to CRISPR screens. The work is of interest, however, many issues need to be addressed.

General remarks

Essential genes are the genes that are critical for survival. Understand essentiality is important to characterize biological systems and to devise strategies to treat cancer. Essential genes have been well characterized for many prokaryotic organisms but not for eukaryotic species. Saccharomyces cerevisiae is the only eukaryotic species, for which systematic essentiality screens have been carried out, revealing that 15-20% of all genes are essential. Human gene essentiality has been studied in cell lines by loss-of-function RNAi screens and, more recently, CRISPR screens. But the major hurdle for these technologies is the lack of specificity. Off-target activities complicate the interpretation of screen results and currently there is not efficient way of evaluating different studies. The work by Hart et al. will help data analysis of lethality screens by providing gold standard reference gene sets of essential/non-essential genes. As Hart et al. said in the manuscript, "a useful yardstick against which cancer functional genomics studies can be measured". Currently, there are only a couple hundred essential genes annotated in the human genome. This study will be a great resource for essential human genes as well as a great step forward for evaluating lethality type screens using RNAi and CRISPRs.

Major issues to be addressed by the authors

1. All the essential gene sets assembled were based on pooled shRNA screens using the TRC library. Since there maybe certain bias and issues associated with each library, it would be informative to see the analysis done with other RNAi libraries.

Our goal in this manuscript was to develop a framework for measuring error rates in genomic perturbation screens. The Reviewer raises an important point regarding library bias, which can only be addressed with comparable datasets generated using a different shRNA library (i.e. not the TRC library). Importantly, no such large-scale datasets exist, and most of the published screens using alternate shRNA or siRNA libraries are not surveying a large set of cell lines (which increase exposure to tissue-specific essentials), or are not genome-scale (e.g. kinome screens).

We have added a paragraph in the main text and a supplementary figure describing the derivation of core essentials from the Achilles data (PMID: 21746896). Importantly, the pooled library is a substantially different set of shRNAs (54k shRNAs in the Achilles screens vs 80k shRNAs in the Marcotte dataset). Integrating the COLT (ie. Marcotte) and Achilles data, and filtering for constitutive expression, yields 360 core essential genes. The population genetics study in Figure 8 is updated to reflect this new definition of core essentials.

2. OTEs have been linked to the mechanism of action of miRNAs, in which the 'seed region', a short sequence of bases 2-8 on the 5' end of the RNAi duplex, is complementary to the 3' untranslated regions (UTRs) of multiple mRNAs, causing degradation of their associated transcripts. Usually this type of OTEs is phenotype specific. Sigoillot et al (A bioinformatics method identifies prominent off-targeted transcripts in RNAi screens. 2012, Nature method) showed that many top hits of the spindle assembly checkpoint screen analyzed have seed region match to the 3'UTR of MAD2, therefore are OTEs. Thus, the authors should go back to the sequences of RNAi reagents of the essential genes identified and perform a seed region analysis.

The Bayes Factor method we describe takes the sum of evidence from all the hairpins targeting a gene. In practice none of the genes we characterize as essential are so classified on the basis of a single hairpin. In deriving a seed set of training essentials, we require that the top half of all hairpins targeting a gene show significant projection on the 1st left singular vector. In calculating Bayes

Factors, we accumulate evidence across all observations of all hairpins targeting a gene. Single hairpin off-target effects are therefore highly unlikely to be a major source of false positives in our results. Moreover, the execution of such a seed region analysis would be problematic, because the definition of a meaningful off-target effect in this context would be finding an essential gene that is unintentionally targeted. But how do we know what these other essential genes are? The purpose of this study is to define that set in the first place.

Nevertheless, we agree that a seed analysis of known or suspected off-target hairpins is an excellent idea as a follow-up to this study. With a defined set of essential genes, suspected off-target hairpins (e.g. those which show strong SVD projections but do not target expressed or known essential genes) would be excellent candidates for this type of analysis.

3. All the essential genes identified in this study were based on the RNAi data of cancer cell lines. The cellular background is quite different between cancer cell lines and normal cell lines, thus the study should be performed on normal cell line data as well.

The Reviewer raises a common question and this is the standard to which most cancer functional genomics studies strive. That is, comparison of “normal” and matched “cancerous” tissue/cells. To specifically address this point, we did include HPDE cells as one of the “normal” cell lines in our dataset. HPDE cells were derived from human pancreatic duct epithelium, have been immortalized with E6/E7 and are non-tumorigenic – they are the gold standard “normal” cell line for cell biology studies relating to pancreatic ductal adenocarcinomas (PMID:9665487). In large datasets, “normal” cell lines typically cluster in with cancer cell lines in terms of essential genes. In fact, the Achilles data contains several “normal” cell lines that cluster in with cancer cell lines in terms of genetic dependencies. All of the published data to date (as well as some of our unpublished data with “normal” breast and ovarian cell lines) suggest that “normal” cells that grow in tissue culture conditions cluster with different cancer genotypes. It is not clear at this point if this is due to tissue of origin and many more screens will need to be carried out with “normal” cells to determine this.

4. Negative gene sets assembled were based only with genes that are not expressed. This set is potentially biased for genes that are not properly annotated or not detected due to technical issues during experiments. Thus, it is important to include genes that are expressed but do not have lethality phenotypes.

We agree that a set of known nonessential expressed genes would be an excellent addition to our training set. Unfortunately no such set exists; the global non-essentiality of a gene is a hard thing to prove. While the non-expressed set may in fact be contaminated by improperly annotated genes, the size of the data set (nearly 1,000 genes) suggests that a small level of contamination will have only marginal effect on our results. Although we appreciate the Reviewer’s comment, we would argue that introducing expressed genes into our non-essential set would run a higher risk of contaminating our results.

5. There are about 120 human essential genes annotated from literature at DEG (Database of Essential Gene, <http://www.essentialgene.org/>). There is not much overlap of DEG gene list with the essential genes sets assembled in this study. It would be great for the authors to provide the explanation of the lack of overlap and why they were not included.

This is an excellent example of the difference between peripheral and core essentials, or between organismal and cell-line essentials. The paper in question (Liao & Zhang, 2008, PMID 18458337) defines essential genes as human null mutations that give rise to juvenile lethality or sterility. As we describe in Figure 8, the organism is not tolerant to null mutations in core essential genes, which would likely result in embryonic lethality. We thank the reviewer for this suggestion, and we have added this note in the main text.

6. Authors are reporting an interesting observation that AGO2 was the top ranked correlation when measuring the correlation between gene expression and screen F-measure, suggesting that AGO2 mRNA level may explain why some screens perform better than others. It would be great if the authors could further confirm this observation by providing the data of AGO2 expression by either western blot or qPCR in several cell lines.

We agree that the observation is interesting, but conducting new experiments to accurately nail down the predictive power of AGO2 expression is an expensive and time-consuming process whose cost is not justified by its benefit for this study. A follow up study in an isogenic cell line with varying AGO2 expression levels might be very informative, but could also stand on its own as an independent study.

7. The authors observed a negative correlation between copy number and essentiality. They argued that "increased copy number yields protein levels in excess of stoichiometric requirements for protein complex function", which may explain false negatives in essential genes in some screens. Copy number does not always reflect the expression level of a gene, so it would be more convincing to show the expression data of these genes.

We are exploring the relationship between gene expression level and sensitivity to RNAi perturbation in an evolutionary context as part of a different study (submitted), and feel it would be inappropriate to publish what would effectively be the same result twice.

Minor issues to be addressed by the authors

1. When the seed reference set of essential genes was assembled, the authors selected the expressed genes based on NGS data from 17 cell lines and 16 BodyMap tissues. The expression threshold used was RPKM = 1. This threshold seems low and needs to be justified.

Figures S2a and S2b show the mean/variance relationship for genes expressed in the Encode and BodyMap samples. R/FPKM = 1 is typically within the main peak of gene expression (see Hart et al., 2012 [PMID 24215113]). In context, the low-variance filter is a much more strict filter than the expression level in defining constitutive genes.

2. The authors compared the human essential genes with the essential genes from mouse KO studies. It would be interesting to compare human essential genes with yeast essential genes as well.

We have added this panel to Figure 4.

3. Typos: On page 15, paragraph 1, figure 6a should be figure 7a. On page 15, paragraph 2, figure 6b should be figure 7b.

We thank the reviewer for catching these typos.

Reviewer #3:

The manuscript by Hart et al. describes a useful method for the analysis of genetic screens both RNA-i based and CRISPR-based.

The authors derive what they call "gold-standard reference sets of essential and non-essential genes", mainly based on the analysis of RNAi screens across multiple cell lines, described previously by the same group in Marcotte et al., 2012. The analysis provides a useful set of Core Essential genes, which are predicted to be essential in most (all) human cells and a broader set of Total Essential, comprising Essential genes which are necessary only in a subset of cell lines. They also identify a set of putative Non-Essential genes.

These gene sets are then used by the authors to re-evaluate every single cell line as an individual screen in the dataset described in Marcotte et al. Interestingly --and somewhat expectedly-- they found that two main sources of variability and of high rate of false positives in the screens are low AGO2 expression and high copy number levels (amplifications) for single genes. These are important aspects when screens are performed in non-isogenic highly aneuploid cell lines.

Below are 5 minor comments and suggestions for potential improvements of the paper. They are in order of appearance in the paper but the most important one is point 3

1) The two sets of Essential and Non Essential genes are at least partially biased for the RNAi library and types of cell lines used in the screens, whose analysis is utilized to derive the gene sets themselves (Marcotte et al paper). Even though the screen-dependent bias should be buffered by the simultaneous analysis of multiple cell lines, it is not possible to exclude the presence of such a bias. Thus, since the authors define these two gene sets "gold-standard reference sets of essential and non-essential genes", they should at least add a note within the text stating the fact that these two lists are subject to potential weakness of the RNAi screen methodology itself, so most likely are not comprehensive and are necessarily dependent on the screen used to derive them. Thus, even though very useful, they cannot represent an absolute "gold standard" but a very good list approaching a gold standard..

This bias may come into play when for example the authors evaluate the performance of the screens described in Cheung et al, 2011 using these gene sets.

A better evaluation of these screens would be using in silico gene sets or after generation of another Essential and Non essential gene sets based on these screens to be evaluated (i.e. repeating the analysis that the authors perform on the screens described by Marcotte et al., 2012 in the first part of the text).

Related to this is the fact that in theory the "gold standard" sets of Essential and Non Essential genes could be improved by including in the analysis all the RNAi screens (Marcotte et al , Cheung et al , Solimini et al.) and potentially CRISPR screens described in the text.

We thank the reviewer for these insightful comments. We have added a summary analysis of the Cheung et al. data and have updated our “gold standard” set to include both the Marcotte et al. and Cheung et al. data. The addition of this data causes a substantial increase in our total number of “global essentials,” from 217 to 360 genes. The discussion highlights the false negative problem inherent to large-scale screens. We are skeptical of including data from smaller studies, however, as it increases the likelihood of capturing tissue-specific essentials.

We don't fully agree with the reviewer's characterization of the “gold standard” label. While our list certainly represents a biased sample of true universally essential genes, we hope that our efforts have minimized the false positives in this group, likely at the cost of additional false negatives. The online Oxford English Dictionary defines “gold standard” as “a thing of superior quality which serves as a point of reference against which other things of its type may be compared.” A functional genomics gold standard need not be complete; to be useful, it need only be highly accurate, and we believe our reference sets meet this criterion.

2) The authors develop a new method for evaluating the screens described in Marcotte et al., 2012, using the newly derived Essential and Non Essential gene sets.

They should compare this method with others standard methods commonly used and previously described.

For example another method of evaluation of a screen is by looking "simply" at the enrichment of an in silico list of Essential genes among the genes that are dropping out in a screen, similar to what was described in Marcotte et al., 2012. It would be good if the authors could evaluate each screen (each cell line) in the same screen dataset of Marcotte et al using this method and compare it with the more sophisticated method that they describe. It would be good to know whether the 48 cell lines performing well using the newly developed method would have been identified also with simpler methods or not.

The “simple” enrichment method has two shortcomings that this study was specifically designed to address. First, the *in silico* list of essential genes is necessarily generated by inference rather than by direct experiment, with all accompanying bias (though also true with the experimental set, which we have tried to minimize as described above). Second, the lack of a nonessential reference removes all information about the quality of inference about genes that are ranked similarly with the essential genes. If those genes contain a high proportion of nonessentials, then the screen is necessarily of poorer quality than a similar screen with a lower proportion of nonessentials, even at the same enrichment level.

An interesting comparison would be the enrichment approach using the *in silico* and experimental lists of essential genes, but that naturally presupposes the existence of the experimental set and introduces circularity if it is used to assess the same datasets from which it was derived.

We have compared our global set of essentials (n=291) with the essentials from Marcotte et al (n=293 current gene IDs). The intersection (n=199) shows a high proportion of constitutive, invariant expression (81%). Of the genes unique to Marcotte et al (n=94), only 42% have constitutive, invariant expression, which shows only marginal enrichment over random expectation and suggests a high proportion of false positives in this set. We have added this comparison to the main text.

3) The third issue concerns the evaluation of the CRISPR screen. The authors say that the screen published by Shalem et al. has a N of genes classified as Essential (by their BF methods) that is higher than the cumulative analysis of all the cell lines screened by RNAi. From this analysis the authors suggest that CRISPR screens have greater sensitivity and report this in the abstract as well. The analysis of the CRISPR screen in comparison with RNAi is an important point but should be improved. It would be important if the authors compared RNAi and CRISPR screens done using the same cell line, the melanoma cell line A375, for which the Shalem et al. paper and a previous paper describe both types of screening methods.

This exact type of comparison was not done in the Shalem et al. paper and given the expansion of CRISPR technology would be of great interest for many researchers.

For this type of analysis it would be good to use both the "gold standard" sets derived by the authors and also a theoretical list of Essential genes.

We agree with the reviewer's comments and have substantially increased the CRISPR analysis section. We compare CRISPR results at varying thresholds to gene expression from the same cell line, showing that the degree to which the screen classifies trace-expression genes as essential is an estimator of its error rate. We conduct an identical analysis in several shRNA screens for which sample-matched RNA-seq data is available, and demonstrate quite conclusively that CRISPR (or at least the one published screen) is both more sensitive and more specific than RNAi. However, we have no data from a matching negative-selection shRNA screen in A375. Though Whittaker et al. (PMID: 23288408) use pooled library shRNA to study synthetic lethality in this cell line, their raw data is not available.

4) It would be great if the authors could provide an automatic tool online for the evaluation of a genetic screen.

As noted above, we have included an archive file containing all the scripts, reference sets, and raw data to calculate BFs for one of the cell lines described in the paper (SK-PC-1).

5) A curiosity: Since the authors describe a correlation between lack of efficacy of shRNA in the detection of an Essential gene and the copy number level of that gene in the cell, I was wondering whether this effect may be even more prominent using the CRISPR technology, since it is based on gene inactivation at the genomic level.

We wonder this as well! We expect that the relationship between copy number (a proxy for expression) and RNAi sensitivity only holds for high expression genes, and speculate that this relationship might not be so expression-dependent in CRISPR screens.

Reviewer #4:

Review: Measuring error rates in genomic perturbation screen: gold standards for human functional genomics by Hart et al.

In this manuscript the authors describe the selection of sets of essential and non-essential genes based on the results of a large number of genome scale shRNA screens in a large panel of cell lines. They propose the use of these essential genes as a standard for quality assessment of functional genomic screens. Selection of a seed set of essential genes is based on singular value decomposition of a large matrix comprising shRNA screen viability data. From this, a median hairpin projection was determined, followed by calculation of an enrichment P-value by a hypergeometric test and corrected Q-values. This initial essential gene set was further filtered using expression analysis. The selected essential and non-essential genes were used to calculate a Bayes Factor score for each

gene in each cell line and subsequently an F-measure for screen performance. Using the group of high performing screens, the authors selected 291 genes scoring as essential in more than half of the cell lines, and considered this a core essential list. With this set, an estimation of false discovery rates is made and used to extend the list to 823 genes with an estimated FDR of 6-11%. These 823 genes are enriched for functional protein complexes involved in core biological processes. This set of genes is further used to indicate a negative correlation between gene copy number and essentiality. Finally, the authors define a set of 217 constitutive core essential genes that are used to evaluate other large scale screens, including shRNA and CRISPR based screens.

In conclusion, this manuscript describes the generation of a reference set of cell line essential and non essential genes (based on shRNA screening results) that can be used for quality assessment of functional genomic screens with the goal to validate new technologies and to ensure the appropriate use, interpretation and conclusion of such efforts.

The work described in this manuscript is of high interest to the functional genomic screening community and addresses an important issue of quality standards for large scale screening projects. However, to ensure broad-readership, the analytical methods should be more clearly described and where possible illustrated with an explanatory figure. It would for example be helpful to present a figure that shows on which parts of the large screening matrix the different stages of the analytical procedure were performed and to what extent the essential gene sets identified with the different methods correlate (overlap) with each other.

We have added a new panel in Fig 6 describing the analytical flow of the paper and the relationship of the various derived gene lists; we hope this alleviates some of the confusion. As noted in the text, the “global essentials” set of 823 contains all of the core 217.

The selection of essential genes is solely based on the results of shRNA screens. Although there is ample attention to possible off-target effects, there is no discussion of the limitation of the technology with respect to insufficient knock-down and consequential false negative rates. This point should be discussed, also in the context of the other gene editing technologies. Furthermore, to further substantiate the relevance of the identified essential genes, the authors show that many of these genes belong to essential complexes. One could imagine that this "prior knowledge" is also incorporated in the selection of core essential genes. This would also allow for an estimation of false negative rates for other functional genomic technologies.

In the same way that we derived “constitutive core essentials” from the Marcotte et al data (PMID: 22585861), we have now derived common essentials from the high-performing subset of the Achilles data and compared the results. The union of the two sets contains 360 genes, ~50% larger than that of either set independently, which gives some indication of the false negative rate of the combination of screen methodology, shRNA library, and analytical approach. We have also added substantially to our analysis of the CRISPR screen and show—quite convincingly, in our view—that CRISPR is more sensitive and less error prone, at least for the single data set published so far.

We agree that an in-depth analysis of what is detected vs. what is missed and how they relate to the encoded protein’s interactions, functions, evolutionary context, expression, and paralogs might be an excellent way to predict the full complement of core essential protein, but this will serve as another full research project and we feel is beyond the scope of this manuscript. For this study, we have used functional and other annotation data to increase the accuracy rather than the scope of our findings.

On page 12, the authors discuss a possible correlation between AGO2 expression levels and screen F-measure. This observation is intriguing but rather weak and potentially biased by cell type of origin. To strengthen this conclusion and drastically improve the utility of the presented work to the screening community, this could be tested experimentally by comparing F-measures of screens done in the same cell line, before and after over-expression of AGO2.

We agree that the observation is interesting, but conducting new screens to nail down how AGO2 expression level predicts RNAi screen performance is an expensive and time-consuming process whose cost is not justified by its benefit for this study. A follow up study in an isogenic cell line with varying AGO2 expression levels might be very informative, however.

At the bottom of page 12, the authors suggest that the use of CRISPR technologies could test the role of copy number variation in false negative rates for shRNA screens. However, this technology may also suffer from the same problem where editing of multiple alleles can be rate limiting for this technology.

We fully agree, and have modified the text accordingly.

On page 19, the authors address a potential explanation for the effect of copy number variation and essential gene sensitivity to RNAi perturbation. They state that this can be explained by the fact that these proteins are part of a protein complex. However, one could argue that also non-complexed proteins with e.g. an enzymatic function can increase in expression above the required level of activity thereby reducing the effects of perturbation by RNAi. The conclusion that the findings described in this paper "extend the CYCLOPS concept to all core essential complexes" is not substantiated by the data provided in this manuscript and should be adjusted accordingly.

We fully agree that the relationship between gene copy number (probably just a proxy for expression) and RNAi sensitivity may extend well beyond the protein complexes. The CYCLOPS concept, that copy loss increases sensitivity, is consistent with our data for protein complexes. That other genes may also show this relationship does not, in our view, undermine the finding; rather, it extends it to yet more potentially useful therapeutic targets, and we hope the research community at large follows up on this hypothesis.

Finally, the title of this manuscript states that a gold standard has been established with this work. I would argue that this is an exaggeration as it builds on the results of shRNA screens with their specific limitations. The 'golden standard' list presented here will likely change if for example performed on a large set of CRISPR screens. It would be more appropriate to change the title and all references of a golden standard to for example ": a tool for standardizing human functional genomic screens".

Undoubtedly the list will change, but that does not undermine our conclusions. As we note in our response to Reviewer #3 above, the online Oxford English Dictionary defines "gold standard" as "a thing of superior quality which serves as a point of reference against which other things of its type may be compared." A functional genomics gold standard need not be complete; to be useful, it need only be highly accurate, and we believe our reference sets meet this criterion.

Minor point:

The legend of Figure SIC mentions projections on the left singular vector where the annotation in the figure states right singular projection. This inconsistency should be corrected.

We thank the reviewer for catching this mistake and have updated the figure legend accordingly.

Acceptance letter

24 April 2014

Thank you again for sending us your revised manuscript. We have now heard back from the two referees who agreed to evaluate your revised study. The referees are now satisfied with the modifications made and I am pleased to inform you that your paper has been accepted for publication.

Thank you very much for submitting your work to Molecular Systems Biology.

Reviewer #1:

My previous remarks have been adequately addressed.

Reviewer #2:

The authors have addressed my comments. I recommend publication.